# Handling of intracellular K⁺ determines voltage dependence of plasmalemmal monoamine transporter function

**Shreyas Bhat[1†], Marco Niello[1†], Klaus Schicker[2], Christian Pifl[3], Harald H Sitte[1], Michael Freissmuth[1], Walter Sandtner[1]\***

[1]Institute of Pharmacology and the Gaston H. Glock Research Laboratories for Exploratory Drug Development, Center of Physiology and Pharmacology, Medical University of Vienna, Vienna, Austria; [2]Division of Neurophysiology and Neuropharmacology, Centre for Physiology and Pharmacology, Medical University of Vienna, Vienna, Austria; [3]Center for Brain Research, Medical University of Vienna, Vienna, Austria

**\*For correspondence:**
walter.sandtner@meduniwien.ac.at

[†]These authors contributed equally to this work

**Competing interests:** The authors declare that no competing interests exist.

**Abstract** The concentrative power of the transporters for dopamine (DAT), norepinephrine (NET), and serotonin (SERT) is thought to be fueled by the transmembrane Na⁺ gradient, but it is conceivable that they can also tap other energy sources, for example, membrane voltage and/or the transmembrane K⁺ gradient. We have addressed this by recording uptake of endogenous substrates or the fluorescent substrate APP⁺(4-(4-dimethylamino)phenyl-1-methylpyridinium) under voltage control in cells expressing DAT, NET, or SERT. We have shown that DAT and NET differ from SERT in intracellular handling of K⁺. In DAT and NET, substrate uptake was voltage-dependent due to the transient nature of intracellular K⁺ binding, which precluded K⁺ antiport. SERT, however, antiports K⁺ and achieves voltage-independent transport. Thus, there is a trade-off between maintaining constant uptake and harvesting membrane potential for concentrative power, which we conclude to occur due to subtle differences in the kinetics of co-substrate ion binding in closely related transporters.

## Introduction

The dopamine transporter (DAT), the norepinephrine transporter (NET), and the serotonin transporter (SERT) are members of the solute carrier six (*SLC6*) family. DAT, NET, and SERT mediate reuptake of released dopamine, norepinephrine, and serotonin, respectively (*Kristensen et al., 2011*). By this action, they terminate monoaminergic signaling and—in concert with the vesicular monoamine transporters—replenish vesicular stores. These transporters are secondary active in nature; they utilize the free energy contained in the transmembrane Na⁺ gradient (established by the Na⁺/K⁺ pump) to drive concentrative monoamine uptake into cells in which they are expressed (*Burtscher et al., 2019*). *SLC6* transporters have been postulated to operate via the alternate access mechanism (*Jardetzky, 1966*): they undergo a closed loop of partial reactions constituting a complete transport cycle (*Rudnick and Sandtner, 2019*). These partial reactions require conformational rearrangements and binding/unbinding of substrate and co-substrate ions. It is gratifying to note that crystal structures obtained from the prokaryotic homolog LeuT, *Drosophila* DAT, and human SERT itself support the general concept of alternate access (*Yamashita et al., 2005*; *Penmatsa et al., 2013*; *Coleman et al., 2016*). These crystal structures also reveal SERT and DAT to be closely related. This is evident from the root mean square deviation, which differs by only approximately 1 Å between the outward-facing structure of human SERT and *Drosophila* DAT. DAT, NET, and SERT also share a rich and partially overlapping pharmacology (*Sitte and Freissmuth, 2015*):

there are many drugs that inhibit transporter function by either blocking or inducing reverse transport. These mechanisms account for therapeutics used for the treatment of neuropsychiatric disorders (major depression, general anxiety disorder, and attention-deficit hyperactivity disorder) and for many illicit drugs that are psychoactive and abused (*Hasenhuetl et al., 2019*; *Niello et al., 2020*).

Despite the similarity in structure and function, the three transporters differ in many more aspects than just ligand recognition: the transport stoichiometry of SERT and DAT/NET is considered to be electroneutral and electrogenic, respectively. The crystal structure of both hSERT and dDAT shows two bound $Na^+$ ions. However, only one $Na^+$ ion is thought to be released on the intracellular side in both the transporters (*Rudnick and Sandtner, 2019*). $Cl^-$, on the other hand, has been shown to play a modulatory role in the transport cycle of SERT and DAT, but $Cl^-$ is not essential for the transport stoichiometry (*Erreger et al., 2008*; *Hasenhuetl et al., 2016*). However, it has long been known that SERT antiports $K_{in}^+$, that is, intracellular $K^+$ (*Rudnick and Nelson, 1978*); for DAT and NET, $K_{in}^+$ is thought to be immaterial (*Gu et al., 1994*; *Gu et al., 1996*; *Sonders et al., 1997*; *Erreger et al., 2008*). If true, only SERT can utilize the chemical potential of the cellular $K^+$ gradient to establish and maintain a substrate gradient. It has, therefore, remained enigmatic, why closely related transporters can differ so fundamentally in their stoichiometry and their kinetic decision points. The effects of $K_{in}^+$, intracellular $Na^+$ ($Na_{in}^+$), and membrane voltage on the transport cycle of SERT have been recently analyzed in great detail (*Hasenhuetl et al., 2016*). However, much less is known on how these factors impinge on the transport cycle of DAT and NET (*Galli et al., 1998*; *Sonders et al., 1997*; *Hoffman et al., 1999*; *Prasad and Amara, 2001*; *Erreger et al., 2008*; *Li et al., 2015*). In this study, we investigated the role of intracellular cations and voltage on substrate transport through DAT, NET, and SERT. To this end, we simultaneously recorded substrate-induced currents and uptake of the fluorescent substrate $APP^+$ (4-(4-dimethylamino)phenyl-1-methyl-pyridinium) into single HEK293 cells expressing DAT, NET, and SERT under voltage control. These measurements were conducted in the whole-cell patch-clamp configuration, which allowed for control of the intra- and extracellular ion composition via the electrode and bath solution, respectively. Our analysis revealed that $K_{in}^+$ did bind to the inward-facing state of DAT and NET, but, in contrast to SERT, $K_{in}^+$ was released prior to the return step from the substrate-free inward- to the substrate-free outward-facing conformations. We also found that substrate uptake by DAT and NET, unlike SERT, was voltage-dependent under physiological ionic gradients. Moreover, the absence of $K_{in}^+$ had no appreciable effect on the transport rate of DAT and NET. The transient nature of $K_{in}^+$ binding was incorporated into a refined kinetic model, which highlighted the differences between SERT and DAT/NET. Notably, this model allows for a unifying description, which attributes all existing functional differences between DAT, NET, and SERT to the difference in the handling of $K_{in}^+$.

## Results

### Single-cell uptake of $APP^+$

We combined advantages of transporter-targeted radiotracer assays and electrophysiology by setting up a system wherein $APP^+$ (*Figure 1A*)-mediated uptake through a single DAT-, NET-, or SERT-expressing HEK293 cell was measured under voltage control (*Figure 1B*, left). *Figure 1B*, right, is a theoretical representative of the two channel recordings; the orange trace represents time-dependent increase in fluorescence (i.e., $APP^+$ accumulation intracellularly), while the red trace represents $APP^+$-induced currents. The scheme in *Figure 1C* is a simplified representation of a transport cycle endured by a sodium and chloride-coupled secondary active transporter. Substrate-induced peak currents are transient in nature and reflect the initial movement of substrate and co-substrates through the membrane electric field (b in *Figure 1B*, right, and highlighted as red in *Figure 1C*). The steady-state current, on the other hand, is indicative of transporters cycling in the physiological forward mode: it persists throughout substrate application (a in *Figure 1B*, right, and highlighted as blue in *Figure 1C*). Accordingly, only substrates can induce steady-state currents while inhibitors cannot. *Figure 1D* is an actual representative trace of the two-channel recordings obtained from control (untransfected) HEK293 cells or from HEK293 cells expressing DAT, NET, or SERT voltage-clamped at −60 mV and exposed to 100 µM $APP^+$. In control HEK293 cells (left-hand set of traces in *Figure 1E*), there was a transient sharp increase and decline in fluorescence as $APP^+$ is washed-in

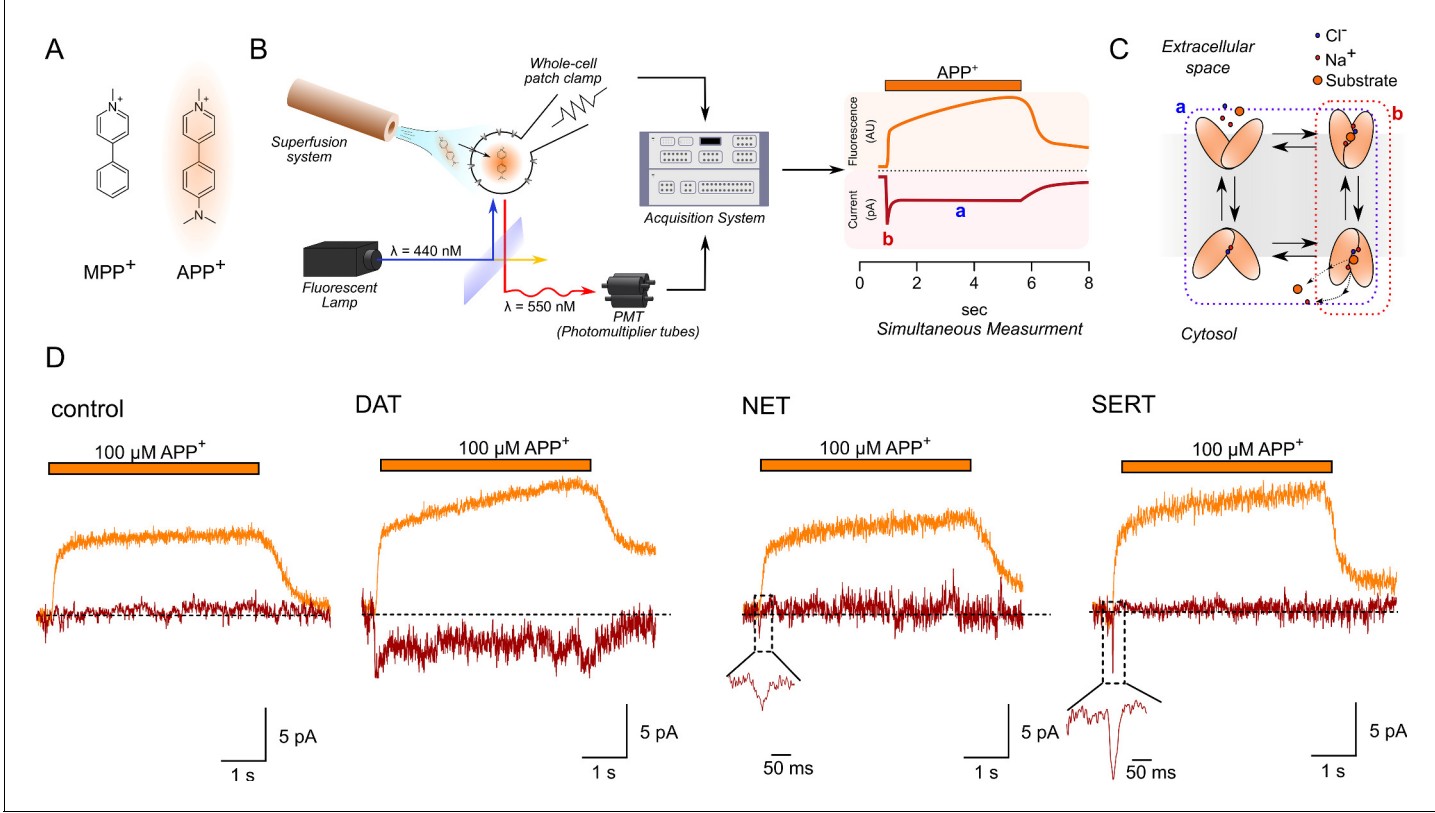

**Figure 1.** Simultaneous measurements of fluorescence and currents. (A) APP$^+$ is a fluorescent analogue of MPP$^+$. Excitation and emission spectra of APP$^+$ occur at wavelengths ranging from 410 to 450 nm and 510 to 550 nm, respectively, which depends on solvent properties. (B) Left, schematics of the setup used in this study. The setup included (1) a dichroic glass that specifically reflects light of 440 nm but allows transmission of those with a wavelength of 550 nm, (2) an inverted microscope with a x100 objective (for single-cell fluorescence recordings), (3) an electrode and patch-clamp amplifier that allows for voltage control, (4) a photomultiplier tube (PMT) that converts emitted fluorescence into electrical signal, and (5) and an acquisition system that allows for filtering, digitizing, and two-channel recordings of APP$^+$-induced fluorescence and currents. Right, a theoretical trace for a two-channel recording that displays simultaneous APP$^+$-induced currents (red trace) and fluorescence (orange trace) in real time. (C) A simplified scheme of a secondary active transporter that employs Na$^+$ and Cl$^-$ as co-substrates. Substrate-induced currents comprise two components: peak current (reactions highlighted in red and b in (B), right) that represents movement of charged substrate/co-substrates through the membrane electric field and steady-state current (reactions highlighted in blue and a in (B), right) that represents transporters functioning in the physiological forward mode. (D) Representative traces of the APP$^+$-induced currents and fluorescence in empty HEK293 cells or in HEK293 cells expressing dopamine transporter (DAT), norepinephrine transporter (NET), and serotonin transporter (SERT) patched under normal physiological ionic conditions. In all traces, the rapid rise and decline in fluorescence on applying and washing off APP$^+$, respectively (see 'Results'), are shown. All three transporters showed a linear increase in fluorescence as a function of time, indicative of APP$^+$ accumulation in cells expressing DAT, SERT, and NET. APP$^+$ induced robust DAT-mediated currents that comprise both peak and steady-state currents. Only peak currents (represented as magnified inset) were seen on APP$^+$ application to cells expressing NET and SERT. AU – arbitrary units.

The online version of this article includes the following figure supplement(s) for figure 1:

**Figure supplement 1.** Epifluorescence recordings of APP+ accumulation.

and washed-out, respectively. There wasn't any concomitant change in current (*Figure 2—figure supplement 1*). In DAT-expressing cells, APP$^+$ induced an inwardly directed current comprised of both a peak and a steady-state component, indicating that APP$^+$ is a DAT substrate. Transport of APP$^+$ into DAT-expressing cells also led to a slow rise in fluorescence. This increase was linear with time and terminated upon removal of APP$^+$. The fluorescence relaxed to a new baseline, which indicated intracellular trapping of APP$^+$. In contrast, APP$^+$ induced only inwardly directed peak currents (but no steady-state currents) in NET- or SERT-expressing cells. While the lack of APP$^+$-induced steady-state currents in SERT and NET may indicate a lack of APP$^+$ transport through these transporters, we, nonetheless, observed a linear increase in APP$^+$ accumulation over time in SERT- or NET-expressing cells. This is indicative of APP$^+$ transport, albeit, poorly by these transporters. The

rapid rise and decline in fluorescence on wash-in and wash-out of APP$^+$ application, seen in control cells, was also observed in DAT-, NET-, and SERT-expressing cells.

The fluorescence intensity of pyridinium dyes such as APP$^+$ is much lower in polar solvents than in their hydrophobic counterparts. For some of these dyes, the quantum yield in the two different environments differed by a factor of 100 (*Fromherz and Heilemann, 1992*). Hence APP$^+$ fluoresces brightly, when adhering to hydrophobic intracellular compartments, but it is barely visible when in the bath solution. This is evident from epifluorescence images of cells expressing DAT upon exposure to APP$^+$ (see *Figure 1—figure supplement 1* and *Videos 1–3*). Importantly, the analysis of these images also aided in our understanding of the nature of the rapid fluorescent component(s). When we integrated the fluorescence intensities in the image over the entire field of view, we observed a robust rapid component. However, this component was much smaller in amplitude when determined for a region of interest encompassing the entire cell. When using a X100 objective, the field of view was approximately 40,000 μm$^2$. Only a fraction (about 1/80$^{th}$) of this surface area was covered by the HEK cell. Thus, although the fluorescence of APP$^+$ in solution was low, its integration over a large area amounted to the rapidly rising and declining component, which we also observed when using the photomultiplier tube (PMT).

## Concentration dependence of APP$^+$-induced currents and fluorescence

The slope of the linear increase in fluorescence has the dimension of a rate (i.e., fluorescence*s$^{-1}$) and is hence a suitable readout for the uptake rate of APP$^+$. We, therefore, determined the concentrations required for achieving half-maximal uptake rates and measured the concomitant currents at −60 mV; original representative traces from single cells expressing DAT, NET, and SERT are shown in panels A and D, B and E, and C and F, respectively, of *Figure 2*. In DAT, the APP$^+$-induced currents increased over the same concentration range as the rise in the rate of fluorescence. Accordingly, the $K_M$ values, which were estimated from fitting the data to a hyperbola ($K_M$ = 27.7 ± 7.1 μM and 21.5 ± 10 μM, respectively), were indistinguishable within experimental error (*Figure 2J*). Dopamine induced transporter-mediated currents (*Figure 2G*) with a $K_M$ of 4.4 ± 1.4 μM. Thus, when compared to dopamine, APP$^+$ is a low-affinity substrate of DAT (*Figure 2J*). In SERT, APP$^+$ did not elicit any appreciable steady-state currents (even at the highest concentration tested, i.e., 600 μM) but a robust concentration-dependent increase in peak current amplitudes (*Figure 2F*). APP$^+$, nonetheless, accumulated intracellularly in SERT-expressing cells (*Figure 2C*), indicating that APP$^+$ was a substrate of SERT, which was translocated inefficiently. The $K_M$ for uptake of APP$^+$ (32.0 ± 13 μM) was about an order of magnitude higher than the $K_M$ of 5-HT (3.6 ± 1.4 μM) estimated from 5HT-induced steady-state currents (*Figure 2L*). This indicates that APP$^+$ uptake is also a low-affinity substrate of SERT. In cells expressing NET, APP$^+$ accumulated in a concentration-dependent manner (*Figure 2B*). Electrophysiological resolution of NET-associated currents revealed that neither norepinephrine (*Figure 2H*) nor APP$^+$ (*Figure 2E*) elicited any steady-state currents in NET on rapid application. In addition, APP$^+$-induced peak currents were considerably smaller than peak currents elicited by norepinephrine (*Figure 2E and H*). These observations can be rationalized by the following hypothetical explanation: (i) NET cycles at rates considerably slower than SERT and DAT (further explored below), thus explaining the lack of substrate-induced steady-state currents and (ii) APP$^+$ is a low-affinity substrate for NET ($K_M$ = 37.3 ± 17 μM; *Figure 2K*), but it

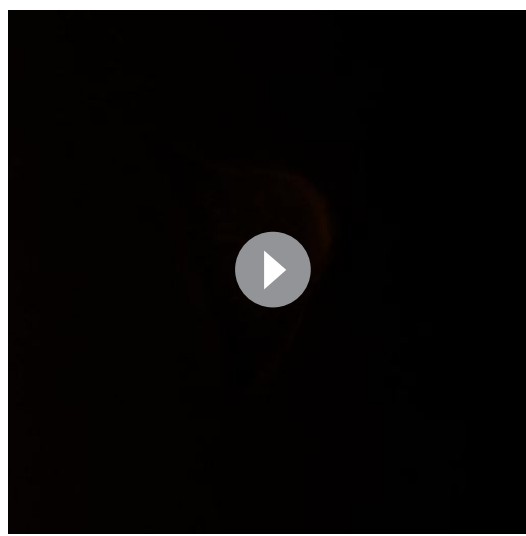

**Video 1.** Accelerated video of APP$^+$ uptake by an HEK293 cell stably expressing DAT (30 μM APP$^+$ was applied at time point 2 s in the video and terminated at time point 4 s in the video). The actual duration of APP$^+$ application was 15 s. The relevant analysis of this video is shown in *Figure 1—figure supplement 1A and B*.

https://elifesciences.org/articles/67996#video1

nevertheless accumulates in NET-expressing cells at appreciable levels, which allows for fluorescence detection. Control untransfected cells did not accumulate APP$^+$ even at the highest concentration tested (600 µM; *Figure 2—figure supplement 1*).

## Voltage dependence of APP$^+$-induced currents and uptake

The data summarized in *Figure 2* indicate that APP$^+$ is a substrate that is taken up with comparable K$_M$ by DAT, NET, and SERT. Accordingly, we applied APP$^+$ for 15 s at a concentration corresponding to the K$_M$ range (30 µM) to examine how changes in voltage (-90 mV to +30 mV) affect APP$^+$ uptake by DAT-, NET-, or SERT-expressing cells in the presence of physiological ionic gradients. It is evident from *Figure 3A* (representative single-cell trace for DAT) and *Figure 3B* (representative single-cell trace for NET) that DAT and NET showed the highest rate of APP$^+$ uptake at -90 mV. The rate of uptake progressively decreased at more positive voltages. In contrast, the change in voltage only had a very modest effect on APP$^+$ uptake by SERT (see *Figure 3C* for a representative single-cell trace for SERT). The slopes acquired over the voltage range were normalized to those observed at -90 mV for each cell and plotted as a function of voltage (circle symbols in *Figure 3J,K and L* for DAT, NET, and SERT, respectively). The plots indicate that uptake of APP$^+$ at +30 mV through DAT, NET, and SERT was ~25, 35, and 80%, respectively, of that at -90 mV. In control cells, changes in voltage did not affect background APP$^+$ binding (*Figure 2—figure supplement 1*). We also assessed the impact of voltage on steady-state currents induced by APP$^+$ (representative traces in *Figure 3D–F*) and of the cognate substrates (representative traces in *Figure 3G–I*). Only in DAT, APP$^+$ and the endogenous substrate evoked transport-associated steady currents (*Figure 3D* and *Figure 3G*, respectively). The voltage dependence of these currents overlaps with that of DAT-mediated APP$^+$ uptake (*Figure 3J*). In SERT, APP$^+$ did not induce sufficiently large steady-state currents to determine any current–voltage relationship (IV) (*Figure 3F*). However, serotonin induced robust steady-state currents (*Figure 3I*), the amplitude of which was reduced by ~50% reduction at +30 mV (diamond symbols, *Figure 3L*). It was not possible to do this comparison in NET (*Figure 3K*), because neither norepinephrine nor APP$^+$ (*Figure 3E and H*) elicited steady-state currents.

## Impact of intracellular cations on APP$^+$ uptake

It is safe to conclude from the observations summarized in *Figure 3* that NET and DAT differ from SERT in their susceptibility to voltage. Transport of APP$^+$ by NET and DAT is voltage-dependent; in contrast, influx of APP$^+$ mediated by SERT is essentially independent of voltage. Previous studies have shown that K$_{in}^+$ was antiported by SERT, but not by NET or DAT (*Rudnick and Nelson, 1978*; *Gu et al., 1996*; *Erreger et al., 2008*). Hence, we surmised that differences in the interaction of intracellular K$^+$ (K$_{in}^+$) with DAT, NET, and SERT may account for the divergent uptake–voltage relation of DAT or NET and of SERT. Accordingly, we varied the intracellular ionic conditions via the patch pipette and compared the rise in APP$^+$ fluorescence over time (represented as AU/sec) in cells expressing DAT, NET, and SERT. The uptake–voltage relationships in the presence of high internal potassium (K$_{in}^+$ = ~163 mM, circle symbols) or in the absence of potassium (NMDG$_{in}^+$ = 163 mM, square symbols) were similar when measured in all three transporters (DAT, p = 0.42; NET, p=0.65; SERT, p=0.63; F-test; *Figure 4A–C*). Owing to the instrumental role of K$_{in}^+$ in the catalytic cycle of SERT, the observed lack of difference in APP$^+$ uptake profiles by SERT-expressing cells in the presence or absence of K$_{in}^+$ seem contradictory. This discrepancy can be explained as follows: (1) SERT can alternatively antiport protons to complete its catalytic cycle (*Keyes and Rudnick, 1982*; *Hasenhuetl et al., 2016*) and (2) APP$^+$ is a poor SERT substrate (as determined by a lack of APP$^+$-induced steady-state currents; *Figures 2F* and *3F*) that may be shuttled into SERT-expressing cells at rates slower than the rate-limiting isomerization of SERT from inward open to outward open state. Unsurprisingly, raising internal sodium to 163 mM (diamond symbols, *Figure 4D–F*) showed significantly reduced uptake by all three transporters when compared to the same profiles in physiological ionic gradients (DAT, p=0.0135; NET, p<0.0001; SERT, p<0.0001 for the y-intercept; F-test; ) this is because high Na$_{in}^+$ precludes substrate dissociation on the intracellular side, thus hampering progression of the physiological transport cycle.

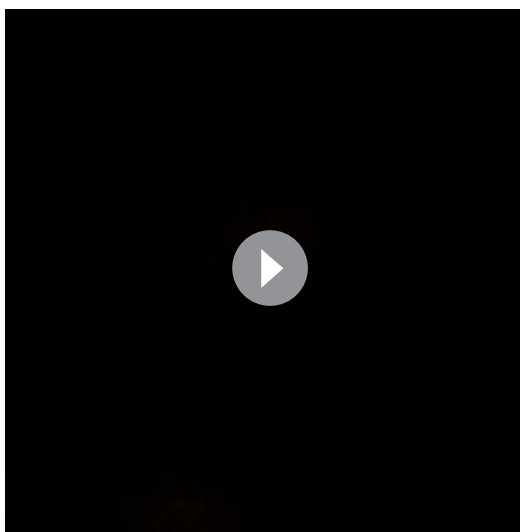

**Video 2.** Accelerated video of APP$^+$ uptake by an HEK293 cell stably expressing DAT (30 µM APP$^+$ was applied at time point 2 s in the video and terminated at time point 4 s in the video). The actual duration of APP$^+$ application was 15 s. The relevant image of this video that highlights intracellular adherence of APP$^+$ is shown in *Figure 1—figure supplement 1C*.
https://elifesciences.org/articles/67996#video2

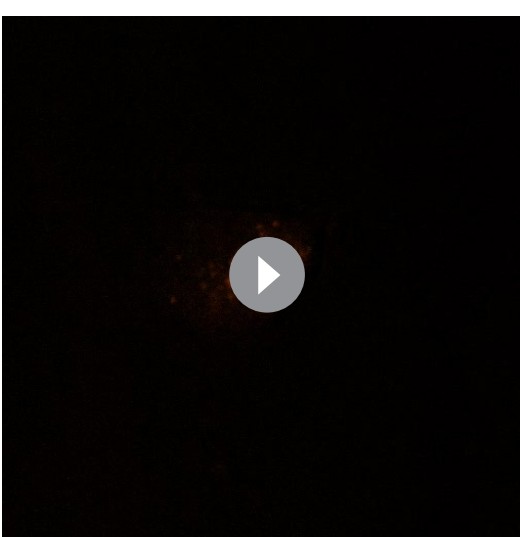

**Video 3.** Accelerated video of a lack of APP$^+$ uptake by an untransfected HEK293 cell (30 µM APP$^+$ was applied at time point 2 s in the video and terminated at time point 4 s in the video). The actual duration of APP$^+$ application was 15 s. The relevant analysis of this video is shown in *Figure 1—figure supplement 1D*.
https://elifesciences.org/articles/67996#video3

## Effect of intracellular cations on currents induced by endogenous substrates

Substrate-induced transporter-mediated currents allow for dissecting partial reactions of the transport cycle (*Erreger et al., 2008*; *Hasenhuetl et al., 2016*). However, APP$^+$ elicited only small currents through SERT and NET. Hence, we relied on endogenous substrates to further analyze the effect of intracellular cations on transporter function. We recorded peak currents through DAT (*Figure 5A,D,G*), NET (*Figure 5B,E,H*), and SERT (*Figure 5C,F,I*), which were elicited by the cognate substrates, in the presence of 163 mM K$_{in}^+$ (*Figure 5A–C*), 163 mM NMDG$_{in}^+$ (*Figure 5D–F*), and 163 mM Na$_{in}^+$ (*Figure 5G–I*) at different voltages. This current reflects translocation of positively charged substrate and Na$^+$ across the plasma membrane, which is hindered at positive potentials. The peak amplitudes obtained over the voltage range tested were normalized to those obtained at −60 mV (please refer to *Figure 5—figure supplement 1* for absolute peak amplitudes in all three transporters and all intracellular conditions tested). In the presence of high intracellular potassium, challenge with cognate neurotransmitters elicited peak current profiles through all three transporters (representative original traces in *Figure 5A–C*) that had linear IV relationships (circle symbols in *Figure 5J–L*). Upon elimination of K$_{in}^+$ (representative original traces in *Figure 5D–E*), this IV relation was significantly steeper as compared to those with high K$_{in}^+$ in DAT and SERT and, to a smaller extent, in NET (DAT and SERT, p<0.0001; NET, p=0.019; F-test, diamond symbols in *Figure 5J–L*). In addition and as previously reported (*Erreger et al., 2008*; *Schicker et al., 2012*), the absence of K$_{in}^+$ eliminated steady-state currents through SERT (*Figure 5C vs F*), but not through DAT (*Figure 5A vs D*).

Because the absence of K$_{in}^+$ affected the slope of the IV relation of the peak current, we surmised that potassium bound from the intracellular side not only to SERT but also possibly to DAT and NET. We explored this conjecture by determining the IV relation of peak currents through all three transporters in the presence of lithium (Li$_{in}^+$ = 163 mM) instead of K$_{in}^+$. Li$^+$ is believed to be an inert cation, because it does not support substrate translocation by *SLC6* transporters. As expected, the IV relation of peak currents through DAT and NET was similar in the presence of 163 mM Li$_{in}^+$ to that recorded in the absence of K$_{in}^+$ (diamond and triangle

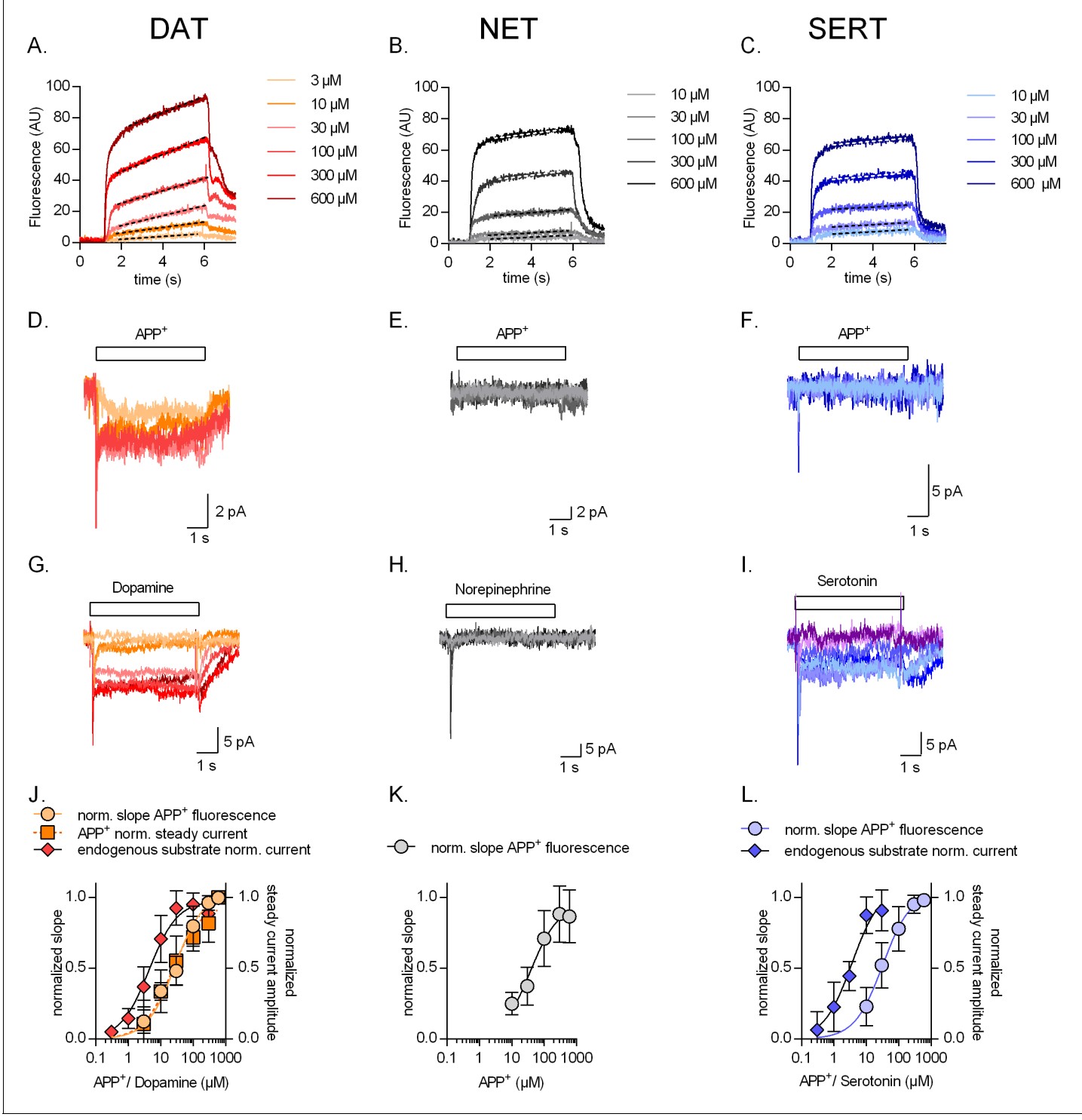

**Figure 2.** Concentration dependence of substrate uptake by monoamine transporters. Representative single-cell traces of concentration-dependent increase in APP[+]-induced fluorescence (A–C) and concomitant currents (D–F) in dopamine transporter (DAT), norepinephrine transporter (NET), and serotonin transporter (SERT), respectively. Representative single-cell traces of currents induced by increasing concentrations of dopamine on DAT (G), norepinephrine on NET (H), and serotonin on SERT (I). (J) Normalized concentration response of APP[+]-induced fluorescence ($K_M$ = 27.7 ± 7.1 µM (n = 11)), APP[+]-induced steady-state currents ($K_M$ = 21.5 ± 10 µM (n = 6)), and dopamine-induced steady-state currents ($K_M$ = 4.4 ± 1.4 µM (n = 6)) in DAT-expressing HEK293 cells. (K) Normalized concentration response of APP[+]-induced fluorescence ($K_M$ = 37.3 ± 17 µM (n = 9)) in HEK293 cells expressing NET. Neither norepinephrine nor APP[+] induced any steady-state currents in NET (even at the highest concentration tested, i.e., 600 µM). (L) Normalized concentration response of APP[+]-induced fluorescence ($K_M$ = 32.0 ± 13 µM (n = 6)) and serotonin-induced steady-state currents ($K_M$ = 3.6 ± 1.4 µM (n = *Figure 2 continued on next page*

*Figure 2 continued*

9)) in SERT-expressing HEK293 cells. APP$^+$ did not induce any steady-state currents in SERT (even at the highest concentration tested, i.e., 600 µM). All experiments were performed under physiological ionic conditions. All fluorescence and current amplitudes were normalized to those obtained at 600 µM (which was set to 1) and the data points were fitted with a rectangular hyperbola. All datasets are represented as means ± SD. AU – arbitrary units, norm. – normalized.

The online version of this article includes the following figure supplement(s) for figure 2:

**Figure supplement 1.** APP+ uptake in control cells.

symbols in *Figure 5J and K*). These observations clearly indicate that K$^+_{in}$ binds to both DAT and NET and rule out an alternative explanation, that is, the effect can be accounted for water and monovalent cations briefly occupying a newly available space in the inner vestibule. SERT, on the other hand, showed shallow IV relations of peak currents with high Li$^+_{in}$ when compared to those acquired in the absence of K$^+_{in}$ (diamond and triangle symbols in *Figure 5L*). This is indicative of Li$^+_{in}$ binding to SERT on the intracellular side. The exact nature of Li$^+_{in}$ binding to SERT has not been reported previously and warrants further investigation. The IV relations of peak currents were similar in the presence of 163 mM K$^+_{in}$ (*Figure 5A–C*) and of 163 mM Na$^+_{in}$ (*Figure 5G–I*) in DAT, NET, and SERT (circle and square symbols in *Figure 5J–L*). This is consistent with the idea that Na$^+_{in}$ and K$^+_{in}$ bind to overlapping sites in these transporters.

## Effect of K$^+_{in}$ on uncoupled conductance and catalytic rate of monoamine transporters

Because internal potassium did not affect DAT-mediated uptake (*Figure 4A*), we examined the role of K$^+_{in}$ in DAT by determining its effect on transport-associated currents. The presence (square symbols in *Figure 6A*) and absence of K$^+_{in}$ (circle symbols in *Figure 6A*) did not change the voltage dependence of the steady-state current. However, the amplitudes of the steady-state currents through DAT were smaller in the absence than in the presence of K$^+_{in}$ (p=0.0334 at -20 mV, n = 9 in each condition; Mann Whitney test). This finding suggests that DAT-mediated currents are not strictly coupled kinetically, that is, a current component exists, which is uncoupled from the transport cycle. Regardless of the intracellular ion composition, steady-state currents were not observed, if NET was challenged with saturating concentrations of norepinephrine (*Figure 6B*). Currents through SERT (amplitudes of which are represented as diamond symbols in *Figure 3L* and square symbols in *Figure 6C*) are completely uncoupled from the catalytic transporter cycle; in spite of its electroneutral stoichiometry, SERT mediates an inwardly directed current, which is eliminated by removal of K$^+_{in}$ (circle symbols in *Figure 6C*). These observations are in line with a previously published work (*Schicker et al., 2012*; *Hasenhuetl et al., 2016*). We further confirmed the contribution of K$^+_{in}$ in the catalytic cycle of all three monoamine transporters by employing a 'two-pulse' peak current recovery protocol (*Hasenhuetl et al., 2016*). This protocol relies on the application of the first pulse (reference) of substrate followed by a variable wash-out interval and the application of a second pulse (test) of the neurotransmitter (representative single-cell traces shown in *Figure 6D*). The first application of substrate results in recruitment of transporters into the transport cycle. Their availability for the second pulse of substrate depends on their completing the transport cycle, that is, on their releasing the substrate on the intracellular side and their subsequent return to the outward-facing conformation. If the intervening wash-out interval is prolonged, a larger fraction of transporters becomes available for the second substrate pulse and this can be gauged from the progressively larger peak amplitudes as a function of time. Thus, the time course of this recovery provides estimates of the catalytic rate of the transporters. As shown in *Figure 6E and F*, the catalytic rates of DAT and NET in the presence or absence of K$^+_{in}$ are very similar. In fact, K$^+_{in}$ fails to render the peak current recovery by DAT voltage-independent (see *Figure 6—figure supplement 1*). In contrast, SERT shows an ~ twofold deceleration in the catalytic rate in the absence of K$^+_{in}$ when compared to its recovery rate in the presence of high K$^+_{in}$ (*Figure 6G*). These observations are in stark contrast to the indistinguishable uptake by SERT of APP$^+$ observed in the presence or absence of K$^+_{in}$ (*Figure 4C*). This discrepancy can be accounted for by APP$^+$ being a poor substrate, an explanation, which is supported by our observations that APP$^+$ did not induce any detectable steady-state currents in SERT (*Figures 2F* and *3F*). All three monoamine transporters can also operate in the

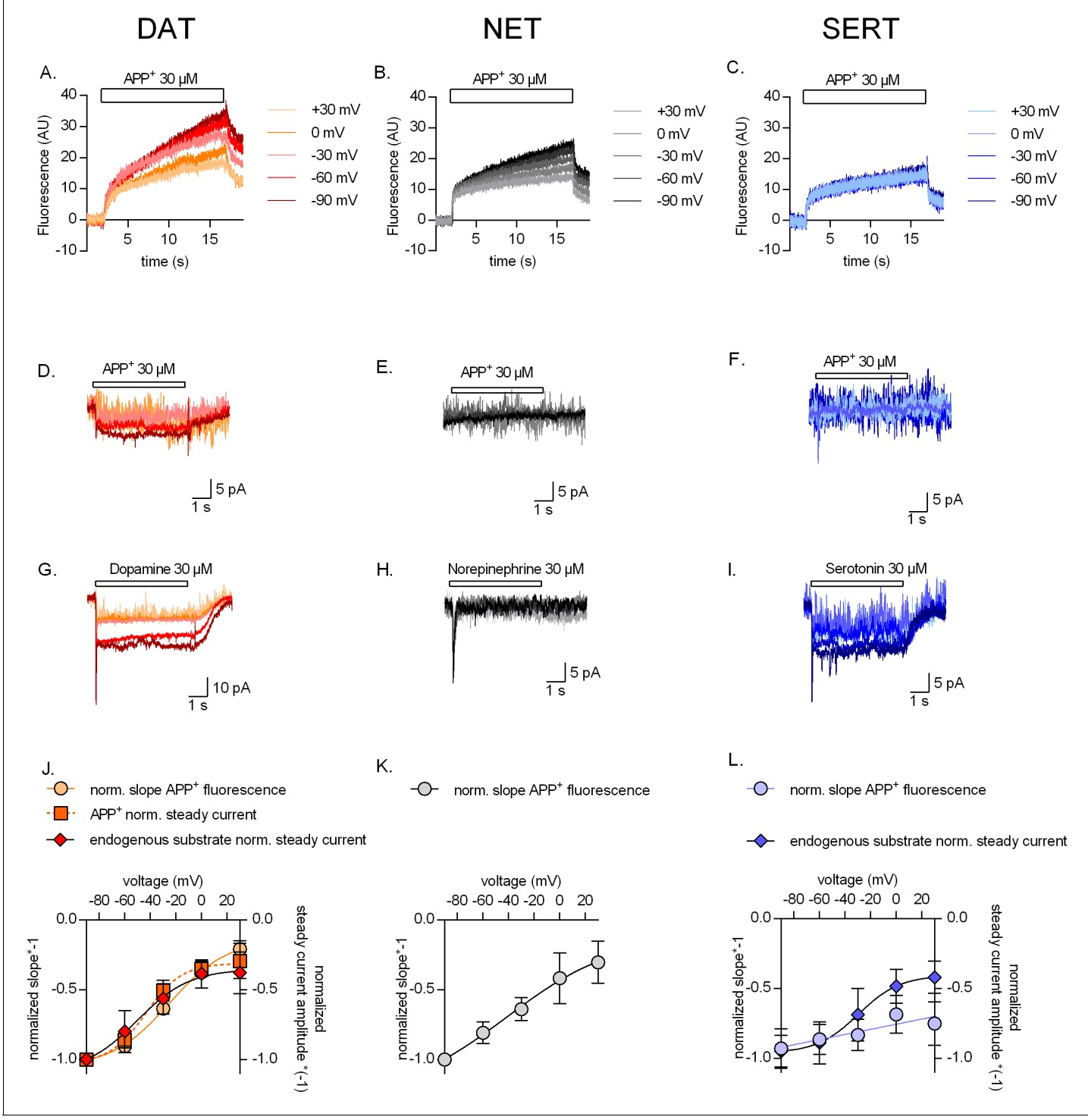

**Figure 3.** Voltage dependence of APP+ uptake by monoamine transporters. Representative single-cell traces of APP$^+$-induced fluorescence (**A–C**) and concomitant currents (**D–F**) under different voltages in dopamine transporter (DAT), norepinephrine transporter (NET), and serotonin transporter (SERT), respectively. APP$^+$ was applied at concentrations of 30 µM. Representative single-cell traces of currents induced by dopamine (30 µM) on DAT (**G**), norepinephrine (30 µM) on NET (**H**), and serotonin (30 µM) on SERT (**I**) under different voltages. (**J**) Normalized voltage dependency of APP$^+$-induced fluorescence (circle symbols, n = 6), APP$^+$-induced steady-state currents (square symbols, n = 6), and dopamine-induced steady-state currents (diamond symbols, n = 6) in DAT-expressing HEK293 cells. (**K**) Normalized voltage dependency of APP$^+$-induced fluorescence (circle symbols, n = 6) in HEK293 cells expressing NET. Neither norepinephrine nor APP$^+$ induced any steady-state currents in NET. (**L**) Normalized voltage dependency of APP$^+$-induced fluorescence (circle symbols, n = 6) and serotonin-induced steady-state currents (diamond symbols, n = 11) in SERT-expressing HEK293 cells. APP$^+$ did

*Figure 3 continued on next page*

*Figure 3 continued*

not induce any steady-state currents in SERT. All experiments were performed under physiological ionic conditions. All fluorescence and current amplitudes were normalized to those obtained at −90 mV (which was set to 1) and the data points were fitted to the Boltzmann equation (except APP$^+$ uptake by SERT, which was fitted to a line). All datasets are represented as means ± SD. AU – arbitrary units, norm. – normalized. We note that the sigmoidal Boltzmann and the line function are both arbitrary fits to the data. Neither one of them are suitable to model the processes, which underlie the depicted voltage dependence. The decision to use one or the other was based on the fidelity of the resulting fit.

exchange mode, which is the basis for the actions of amphetamines (*Sitte and Freissmuth, 2015*). As a control, we examined the recovery in the presence of high Na$_{in}^+$, which precludes cycling in the forward transport mode and thus forces the transporters into exchange: as predicted, high Na$_{in}^+$ accelerated the recovery of all three transporters (square symbols in *Figure 6E–G*).

## A kinetic model for the transport cycle of monoamine transporters

The data, represented in *Figure 6*, can be explained by a model that posits that all monoamine transporters can bind K$_{in}^+$, but that the bound K$_{in}^+$ is released on the intracellular side prior to the return step by DAT and NET. In contrast, K$_{in}^+$ is released on the extracellular side after being antiported by SERT. We tested the plausibility of this hypothesis by resorting to kinetic modeling. As a starting point for modeling DAT and NET, we used the previously proposed kinetic model for DAT by *Erreger et al., 2008*, shaded in grey in *Figure 7A*, which is nested within our proposed model. For NET, we posited a much slower dissociation rate for the substrate (indicated as green text) to account for the small substrate turnover rate and the absence of the steady current component (*Figure 2H* and *Figure 6E*). The model was expanded to account for transient binding of K$_{in}^+$ to DAT and NET. This was achieved by subdividing this event into two consecutive reactions: in the first reaction (when viewed in the clockwise direction), DAT/NET adopts an inward-facing conformation on the trajectory to the occluded state (ToccClK), to which K$_{in}^+$ can still bind, but with reduced affinity; in the second reaction, DAT/NET fully occludes after shedding off K$_{in}^+$(i.e., ToccCl) and rearranges to adopt the outward-facing conformation. We note that conversion of a high- to a low-affinity binding site requires input of free energy. The transport cycle is a multistep process, which would not proceed in the forward direction, if the free energy of the entire process (i.e., ΔG) is positive. To test whether transient K$_{in}^+$ binding to DAT/NET, as proposed in our model, hampers the overall progression of the transport cycle, we mapped out the Gibbs free energy of the outer loop of the kinetic model of DAT (i.e., this path describes the conformational trajectory, which the transporter takes in the presence of K$_{in}^+$ -see scheme in *Figure 7A*) at 0 mV and −60 mV. As seen in *Figure 7—figure supplement 1*, ΔG of the overall process is negative at both potentials despite the high energy cost of transient K$_{in}^+$ binding to DAT. Determination of the exact nature of the energetic requirements of the individual partial reaction is beyond the scope of the study.

In the model of SERT (originally proposed by *Schicker et al., 2012*; *Figure 7B*), we assumed that K$_{in}^+$ was antiported on return of the substrate-free transporter to the outward-facing conformation. In both models (i.e., DAT/NET and SERT), we accounted for the uncoupled current component observed in the presence of K$_{in}^+$ by adding a conducting state, which was in equilibrium with the K$_{in}^+$-bound inward-facing conformation (Tcond, indicated in blue). Notably, voltage-independent substrate transport by SERT in the absence of K$_{in}^+$ (*Figure 4C*) was accounted in the model by the ability of SERT to alternatively antiport a proton (*Keyes and Rudnick, 1982*; *Hasenhuetl et al., 2016*). However, varying intracellular protons did not alter the voltage dependence of the peak current amplitude in DAT (*Figure 5—figure supplement 2*). This finding is consistent with the concept that only SERT can utilize protons in the return step.

We interrogated the kinetic models to generate synthetic data for APP$^+$ transport by DAT (*Figure 7C*), NET (*Figure 7D*), and SERT (*Figure 7E*) at different membrane voltages. The synthetic data generated through the respective kinetic models could faithfully reproduce our experimental findings: (i) only APP$^+$ uptake by SERT was voltage-independent (*Figure 7C–E* and *Figure 3A–C*); (ii) the removal of K$_{in}^+$ abrogated the steady-state current only in SERT but not in DAT (*Figure 7F, G* and *Figure 6A, C*); (iii) the removal of K$_{in}^+$ did not slow down the return of DAT and NET from the inward- to the outward-facing conformation, while it reduced this rate in SERT by twofold (*Figure 6E–G* to *Figure 7—figure supplement 2*). For other simulated datasets, please refer to *Figure 7—figure supplement 2*. This indicates that the underlying assumptions are valid and allow for

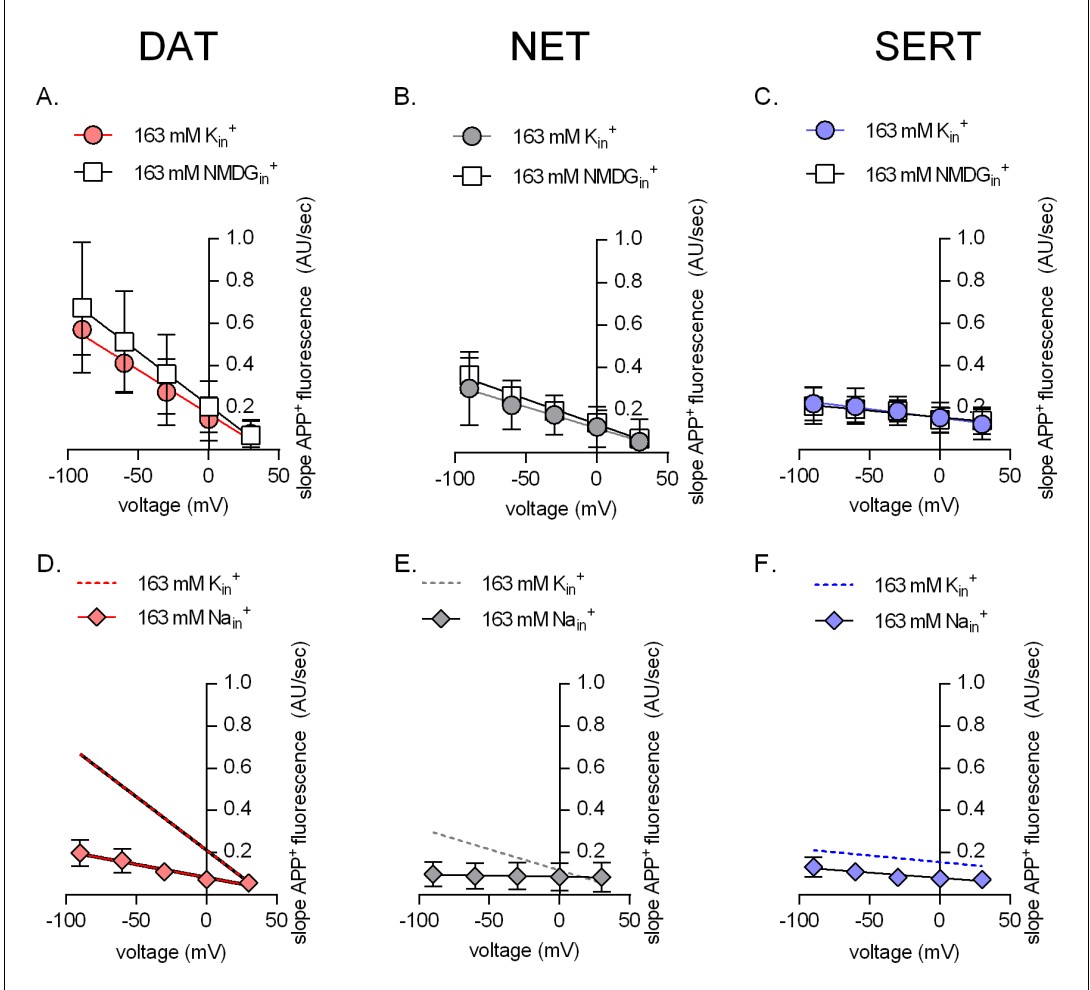

**Figure 4.** The effect of intracellular cations on transporter mediated APP+ uptake. Rate of APP$^+$-associated fluorescent uptake (total rise in absolute fluorescence/unit time – AU/sec) measured in dopamine transporter (DAT) (**A and D**), norepinephrine transporter (NET) (**B and E**), and serotonin transporter (SERT) (**C and F**) under different intracellular conditions and different voltages. Pipette solutions include the physiological intracellular ionic conditions (163 mM $K_{in}^+$, circle symbols), an intracellular condition devoid of $Na_{in}^+$ and $K_{in}^+$ (163 mM $NMDG_{in}^+$, square symbols), or an intracellular environment of high $Na^+$ (163 mM $Na_{in}^+$, diamond symbols). All data points were fitted to the line equation. The dashed lines in (**D–F**) are the same as those obtained by fitting data points obtained from 163 mM $K_{in}^+$ from (**A–C**) for the respective transporters. The slope of the lines in the presence of either 163 mM $NMDG_{in}^+$ or 163 mM $K_{in}^+$ in (**A**) (DAT), (**B**) (NET), and (**C**) (SERT) was not significantly different (F-test). The p-values were 0.42, 0.65, and 0.63, respectively. This suggest that intracellular $K_{in}^+$ does not affect the voltage dependence of APP$^+$ uptake of either of the three monoamine transporters. The slopes of the lines, in the absence and presence of 163 mM $Na_{in}^+$, in (**D**) (DAT, p=0.0135) and (**E**) (NET, p<0.0001) were different (F-test). In (**F**) (SERT), the slope was not significantly different (p=0.66), while the y-intercept was (p<0.0001). This analysis is consistent with the idea that intracellular $Na^+$ hampers APP$^+$ through all three transporters. All datasets are represented as means ± SD; the number of experiments in each individual dataset was six; AU – arbitrary units.

a reasonable approximation, which has explanatory power: the differences in handling of $K_{in}^+$ incorporated into the model are necessary and sufficient to account for the differences in the forward transport mode of DAT, NET, and SERT. Accordingly, while we do not claim that the mechanism that we propose is unique in its ability to explain the data, we argue that it is plausible and parsimonious.

## Discussion

Fluorometric analyses of the SERT, DAT, and NET transport cycle provide mechanistic and functional insights into the modus operandi of these transporters by combining advantages of assays that employ radiolabeled ligands (used to determine global transporter turnover rates) with those that

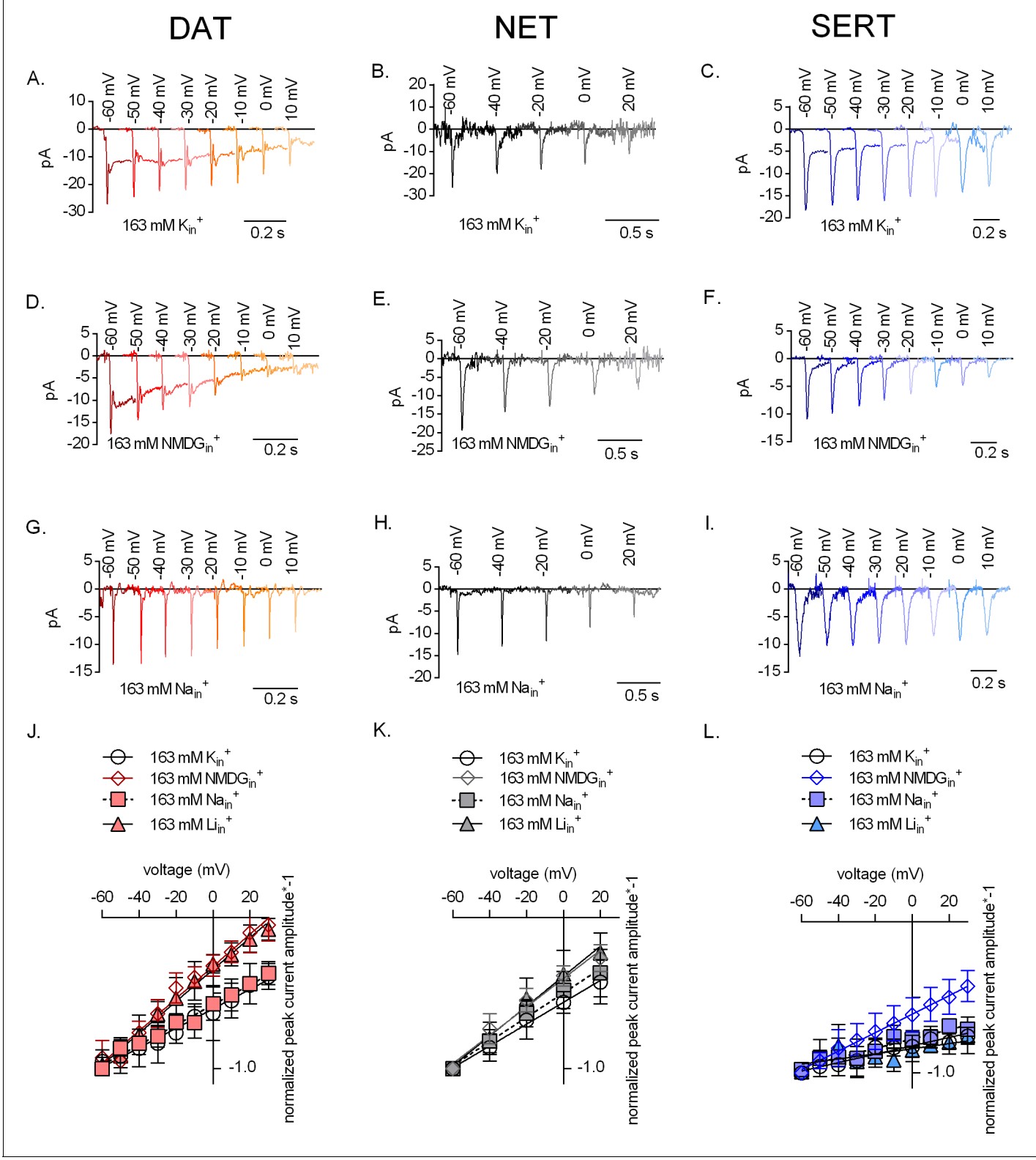

**Figure 5.** The effect of intracellular cations on the voltage dependence of monoamine induced peak currents. Representative single-cell traces of current profiles elicited by 30 μM dopamine in dopamine transporter (DAT) (**A, D, and G**), 30 μM norepinephrine in norepinephrine transporter (NET) (**B, E, and H**), and 30 μM serotonin in serotonin transporter (SERT) (**C, F, and I**) under different intracellular conditions and different voltages. Peak current–voltage relationships of DAT (**J**), NET (**K**), and SERT (**L**) under different pipette solutions that include the physiological intracellular ionic

*Figure 5 continued on next page*

Figure 5 continued

conditions (163 mM $K^+_{in}$, circle symbols), an intracellular condition devoid of $Na^+_{in}$ and $K^+_{in}$ (163 mM $NMDG^+_{in}$, diamond symbols), an intracellular environment of high $Na^+$ (163 mM $Na^+_{in}$, square symbols), and high intracellular $Li^+$ (163 mM $Li^+_{in}$, triangle symbols). Peak current amplitudes were normalized to those obtained at −60 mV (which was set to 1) and the individual datasets were fitted to the line equation. The slope of the voltage dependence in the presence and absence of 163 mM $K^+_{in}$ was significantly different for all three transporters (DAT ($p<0.0001$), NET ($p=0.019$), and SERT ($p<0.0001$); F-test). All data points are represented as means ± SD.

The online version of this article includes the following figure supplement(s) for figure 5:

**Figure supplement 1.** Absolute values of the peak amplitudes elicited in DAT, NET, and SERT.

**Figure supplement 2.** Intracellular protons cannot replace $K^+_{in}$ in DAT.

involve electrophysiology (which provide voltage control and unmatched temporal resolution) (*Schwartz et al., 2006*). Fluorescent substrates of monoamine transporters have previously been used to monitor transporter-mediated uptake in real time (*Schwartz et al., 2003*; *Mason et al., 2005*; *Schwartz et al., 2005*; *Oz et al., 2010*; *Solis et al., 2012*; *Karpowicz et al., 2013*; *Wilson et al., 2014*; *Zwartsen et al., 2017*; *Cao et al., 2020*). One such fluorescent substrate is $APP^+$, a fluorescent analog of $MPP^+$, a neurotoxin that targets monoaminergic neurons (*Javitch et al., 1985*; *Pifl et al., 1991*). Like $MPP^+$, $APP^+$ is also taken up by cells expressing DAT, NET, and SERT; its fluorescent properties are well understood (*Solis et al., 2012*; *Karpowicz et al., 2013*; *Wilson et al., 2014*). In the present study, we relied on $APP^+$ to explore the transport cycle of DAT, NET, or SERT by single-cell analysis, which allowed for simultaneously recording cellular uptake by fluorescence and substrate-induced currents. It was also possible to control the concentrations of the relevant ions and the membrane potential with the unprecedented precision of the whole-cell patch-clamp configuration and to thereby examine their impact on transport rates. To the best of our knowledge, our experiments are the first to address the following question: why do the structurally similar DAT, NET, and SERT differ in transport kinetics and handling of co-substrate binding? It is evident that DAT and NET resemble SERT in most aspects. We show here that all major differences can be accounted for by the distinct handling of $K^+_{in}$. (i) in SERT, physiological $K^+_{in}$ concentrations accelerated the rate of substrate uptake; it was twofold faster than in the absence $K^+_{in}$ (*Figure 6G*). In contrast, DAT and NET returned to the outward-facing state with the same rate regardless of whether $K^+_{in}$ was present or not (*Figure 6E and F*). Accordingly, $K^+_{in}$ did not affect the rate of substrate uptake by DAT and NET (*Figure 4A and B*). (ii) The catalytic rate of SERT was independent of voltage in the presence of physiological ionic gradients (*Figure 3L* and *Figure 5L*). This was not the case for DAT and NET (*Figure 3J and K* and *Figure 5J and K*, respectively). (iii) In all three transporters, release of $Na^+_{in}$ from the inward-facing conformation was electrogenic. In SERT, this electrogenic $Na^+_{in}$ dissociation was canceled out by electrogenic $K^+_{in}$ binding to the inward-open empty transporter and its subsequent antiport, thereby rendering the cycle completion rate voltage-independent. In DAT and NET, however, the cycle completion rate remained voltage-dependent despite the fact that $K^+_{in}$ also bound in a voltage-dependent manner. (iv) $K^+_{in}$ is also relevant to account for the distinct nature of the steady-state current component in SERT and DAT. The steady current component carried by the electroneutral SERT was produced by an uncoupled $Na^+$ flux through a channel state that was in equilibrium with the $K^+_{in}$-bound inward-facing conformation (*Schicker et al., 2012*). This uncoupling explains the existence of differences in the voltage dependence of SERT-mediated uptake and of steady-state currents (*Figure 3L*). DAT-mediated transport was accompanied by the translocation of coupled net positive charges in each cycle. Thus, DAT-mediated steady-state currents were originally modeled to be strictly coupled to substrate transport (*Erreger et al., 2008*). However, our data suggest that DAT also carries a previously suggested uncoupled current component (*Sonders et al., 1997*; *Sitte et al., 1998*; *Carvelli et al., 2004*), which adds to those associated with DAT-mediated ionic transport. This uncoupled current in DAT, just like in SERT, is contingent on the presence of intracellular $K^+_{in}$.

The binding site for $K^+_{in}$ in DAT, NET, and SERT is largely unknown. Binding of $K^+_{in}$ to DAT at the $Na_2$ site was proposed earlier by a study that employed extensive molecular dynamic simulations to understand intracellular $Na^+$ dissociation from DAT (*Razavi et al., 2017*). The amino acids that coordinate and form the $Na_2$ site are absolutely conserved in all three plasmalemmal monoamine transporters (*Penmatsa et al., 2013*; *Coleman et al., 2016*; *Cheng and Bahar, 2019*). In SERT, $Na^+$ binding to $Na_2$ site is thought to trigger conformational transitions essential for physiological substrate

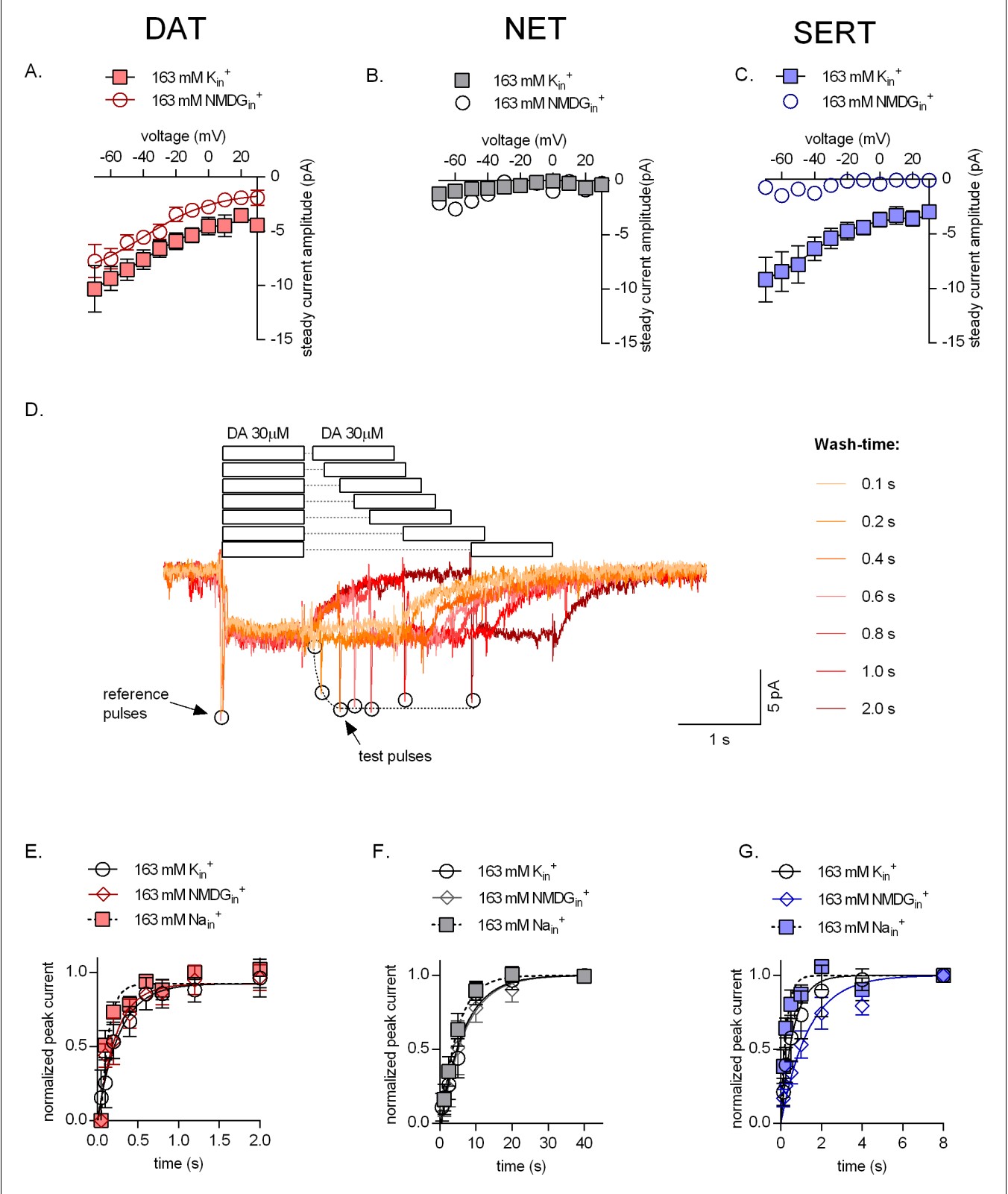

**Figure 6.** Effect of intracellular potassium on the catalytic rate of monoamine transporters. (A, B, C) Amplitudes of steady-state currents elicited by 30 μM dopamine on dopamine transporter (DAT) and 30 μM norepinephrine on norepinephrine transporter (NET), and 30 μM serotonin on serotonin transporter (SERT), respectively, as a function of voltage under physiological intracellular conditions (square symbols) and an intracellular environment devoid of $Na^+$ and $K^+$ (163 mM $NMDG_{in}^+$, circle symbols). NET did not elicit any steady-state currents on exposure to norepinephrine. The data points in

*Figure 6 continued on next page*

**Figure 6 continued**

(A), (B), and (C) are represented as means ± SEM. The lines indicate fits of the Boltzmann equation to the data points. For DAT (A), we compared the amplitude of the current in the presence and absence of 163 mM $K_{in}^+$ at -20 mV. The difference in amplitude at this potential was significant (p=0.0334); Mann Whitney test (n = 9; each). (D) Representative single-cell trace of the two-pulse protocol: 30 μM dopamine was applied to a DAT-expressing cell to generate a reference pulse followed by variable wash times and a repeated pulse of 30 μM dopamine again (test pulse). The peak current amplitudes (which represent available binding sites on completion of the transport cycle) are plotted as a function of time to determine transporter catalytic rate. Catalytic rates determined for DAT (E), NET (F), and SERT (G) under intracellular conditions that are physiological (163 mM $K_{in}^+$, circle symbols), devoid of intracellular $Na^+$ or $K^+$ (163 mM $NMDG_{in}^+$, diamond symbols) or contain high intracellular $Na^+$ (163 mM $Na_{in}^+$, square symbols). Peak current amplitudes obtained at each test pulse were normalized to those of the reference peak (which was set to 1) and fitted with a mono-exponential function. The catalytic rates obtained were as follows: DAT (163 mM $K_{in}^+$ - 3.74 ± 0.76 s$^{-1}$; 163 mM $Na_{in}^+$ - 10.68 ± 2.14 s$^{-1}$; 163 mM $NMDG_{in}^+$ - 4.45 ± 0.93 s$^{-1}$), NET (163 mM $K_{in}^+$ - 0.15 ± 0.013 s$^{-1}$; 163 mM $Na_{in}^+$ - 0.22 ± 0.016 s$^{-1}$; 163 mM $NMDG_{in}^+$ - 0.15 ± 0.014 s$^{-1}$), and SERT (163 mM $K_{in}^+$ - 1.61 ± 0.12 s$^{-1}$; 163 mM $Na_{in}^+$ - 2.57 ± 0.28 s$^{-1}$; 163 mM $NMDG_{in}^+$ - 0.74 ± 0.058 s$^{-1}$). The data in (E), (F), and (G) are represented as means ± SD (n = 5 for each condition).

The online version of this article includes the following figure supplement(s) for figure 6:

**Figure supplement 1.** Intracellular $K^+$ does not render the time course of peak current recovery of DAT voltage independent.

transport (*Felts et al., 2014*). Dissociation of $Na^+$ from the $Na_2$ site, in turn, leads to water flux from the cytosol that promotes transition of the transporter into the inward-facing conformation and subsequent substrate dissociation on the intracellular side (*Cheng and Bahar, 2019*). Our results highlight the fact that the voltage dependence of peak amplitudes is identical in the presence of high $Na_{in}^+$ and high $K_{in}^+$ (*Figure 5J–L*) in all three plasmalemmal monoamine transporters; these observations support the conjecture that $K_{in}^+$ and $Na_{in}^+$ binding occurs at the $Na_2$ site in a mutually exclusive manner. Interestingly, differences between DAT/NET and SERT are further substantiated by the ability of SERT to bind to intracellular $Li^+$. The exact nature of this interaction is unknown and necessitates an in-depth investigation that is beyond the scope of this study.

The most parsimonious explanation for all differences between SERT, NET, and DAT was to posit that $K_{in}^+$ is antiported by SERT but not by DAT and NET. All three transporters carry a negative charge through the membrane on return from the substrate-free inward- to the substrate-free outward-facing conformation (presumably a negatively charged amino-acid). In the case of SERT, however, the charge on the transporter is neutralized by the counter-transported $K_{in}^+$. Because the return step is slow and therefore rate-limiting, it determines the voltage dependence of substrate uptake. We note that the lack of voltage dependence, which we observed in the SERT-mediated uptake, is not contingent on its assumed electroneutrality. Even an electroneutral transporter can support voltage-dependent uptake if the rate-limiting step is associated with charge movement. The $K_{in}^+$-binding site in SERT, alternatively, can also accept protons (*Keyes and Rudnick, 1982*). Hence protons—as an alternative co-substrate that is antiported—support the return step from the inward- to the outward-facing substrate-free conformation (*Hasenhuetl et al., 2016*). The alternative is to postulate, based on recent evidence in LeuT (*Billesbølle et al., 2016*), that antiport of $K_{in}^+$ is a general feature of all *SLC6* transporters. However, this can be refuted for DAT and NET for the following reasons: the presence or absence of $K_{in}^+$ did not change their catalytic rates (*Figure 6E and F*). Thus, in the absence of $K_{in}^+$, the transporters seem to return from the substrate-free inward- to the substrate-free outward-facing conformation. In this case, however, the transporters carried one positive charge less through the membrane. This change in ion translocation must, therefore, translate into a concomitant, substantial change in the voltage dependence of substrate transport, that is, the voltage dependence of transport ought to be much steeper in the absence than in the presence of $K_{in}^+$. This was not observed (*Figure 4A and B*). Additionally, $H^+$ failed to accelerate the catalytic rate of DAT (data not shown) and the slope of the IV curve for the peak current remained steep (*Figure 5—figure supplement 2*). These observations indicate that protons (like $K_{in}^+$) cannot be antiported by DAT.

In monoamine transporters, there is a continuum between full substrates, partial substrates, atypical inhibitors, and typical inhibitors (*Hasenhuetl et al., 2019*; *Bhat et al., 2019*). Interestingly, $APP^+$ is a full substrate of DAT: the currents, which were elicited by $APP^+$, were indistinguishable to those induced by the cognate substrate dopamine and other full substrates such as D-amphetamine (*Erreger et al., 2008*). In contrast, $APP^+$ elicited the peak current but failed to induce the steady current through SERT, which was readily seen in the presence of 5-HT. In oocytes expressing SERT, $APP^+$-elicited currents reached only ~20% of the amplitudes of the 5-HT-induced currents

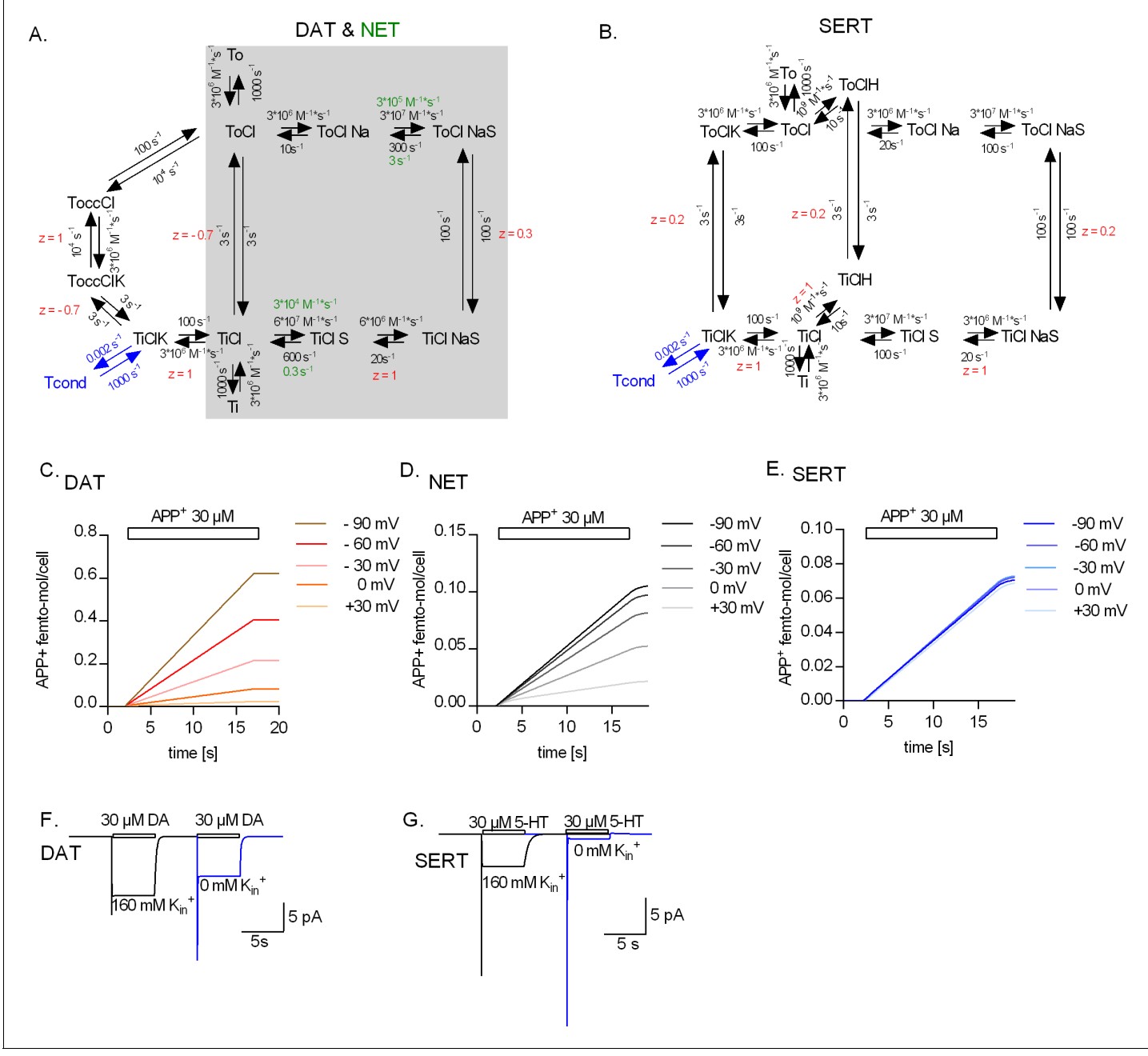

**Figure 7.** Kinetic models of the catalytic cycle of the monoamine transporters. (**A**) Reaction scheme of dopamine transporter (DAT) and norepinephrine transporter (NET). Shaded in grey is the original scheme proposed by *Erreger et al., 2008*. The original scheme is nested in the refined model. We assumed that DAT and NET not only share the same scheme but also most parameters. However, to account for the smaller turn-over rate of NET and the absence (or a lack of detection) of the steady current component on challenge with norepinephrine, we posited slower substrate-binding kinetics for NET (rate constants indicated in green). (**B**) Reaction scheme of serotonin transporter (SERT). Simulated APP$^+$ uptake through DAT (**C**), NET (**D**), and SERT (**E**). $k_{on}$ and $k_{off}$ for APP$^+$ were set to $9*10^5$ s$^{-1}*$M$^{-1}$ and 30 s$^{-1}$, $3*10^5$ s$^{-1}*$M$^{-1}$ and 1 s$^{-1}$, and $1*10^5$ s$^{-1}*$M$^{-1}$ and 1 s$^{-1}$ for DAT, NET, and SERT, respectively. Simulated substrate-induced currents for DAT (**F**) and SERT (**G**) in the presence (black trace) and absence (blue trace) of 163 mM K$^+_{in}$. The current in the presence of 163 mM K$^+_{in}$ is the sum of the coupled and uncoupled currents. The current in the absence of K$^+_{in}$ is the coupled current in isolation. In simulations (**F**) and (**G**), we assumed a membrane voltage of -60 mV.

The online version of this article includes the following figure supplement(s) for figure 7:

**Figure supplement 1.** Model prediction of the energy landscape of the transport cycle endured by DAT.

**Figure supplement 2.** K$^+_{in}$ binding to inward-facing state of DAT affects the voltage dependence of the peak current.

(*Solis et al., 2012*). In SERT-expressing HEK293 cells, the cognate substrate elicits currents of a magnitude in the low pA range. Hence, transport-associated currents induced by APP$^+$ are expected to be lost in the noise. Taken together these observations, APP$^+$ is a poor substrate for SERT: its actions can be rationalized by assuming that it traps transporters in one or several conformational intermediates, which are exited with a rate slower than the return step. Therefore, DAT and SERT diverge in their handling of APP$^+$: while APP$^+$ had similar $K_M$ values for DAT and SERT, the catalytic rate of the transport cycle was equivalent to that of the cognate substrate in DAT, but substantially lower in SERT.

Originally, NET expressed in HEK293 cells was reported to support both a peak and a steady current, when challenged with a substrate (*Galli et al., 1995*; *Sommerauer et al., 2012*). However, in the present study, we only observed the peak current with our superfusion system, which allowed for a rapid exchange of solutions: neither APP$^+$ nor the cognate substrate norepinephrine elicited a steady current. The absence of the steady current can be attributed to the very slow catalytic rate of NET: it is evident that NET, in contrast to DAT and SERT, returns on a timescale of seconds from the inward- to the outward-facing state. NET dwells in the inward-facing state with a lifetime of $\tau = {\sim}7$ s (*Figure 6F*), which is >10–20 times longer than the dwell time of DAT and SERT (DAT, $\tau = {\sim}0.3$ s; SERT, $\tau = {\sim}0.6$ s; *Figure 6E and G*). Turnover numbers determined by maximal uptake rates divided by maximal binding sites were also reported as five times lower in NET than in DAT expressed in COS-7 and human neuroblastoma cells (*Pifl et al., 1996*). Thus, this very slow turnover explains the absence of coupled or uncoupled NET-mediated currents in spite of the proposed electrogenic stoichiometry (*Gu et al., 1996*).

Our analysis provides a unifying concept of substrate transport through all three monoamine transporters, that is, they are equivalent in all aspects of their transport cycle but one: in SERT, the binding site for $K_{in}^+$ remains intact upon conversion of the transporter from the inward- to the outward-facing conformation. In contrast, this binding site is less stable in DAT and NET. The resulting loss in affinity leads to the shedding of $K_{in}^+$ prior to the return step. The energetics associated with this change in affinity has little to no bearing on the overall spontaneity of forward substrate transport in DAT (*Figure 7—figure supplement 1*). The repercussions of this subtle difference are profound: SERT and DAT/NET differ (i) in their voltage dependence of substrate uptake, (ii) in the nature of the substrate-induced current, and (iii) in the energy sources tapped for concentrative substrate transport. If DAT and NET do not antiport $K_{in}^+$, its concentrative power must be independent of the existing $K^+$ gradient across the plasma membrane. On the other hand, if the stoichiometry of DAT and NET is electrogenic, a change in membrane voltage is predicted to increase or decrease substrate uptake at steady state depending on the direction of the voltage change. In this context, it is important to note that the experiments conducted in the present study all report on substrate transport at pre-steady state. Additional insights on whether or not DAT/NET can antiport $K_{in}^+$ can come from experiments conducted at the thermodynamic equilibrium. Such experiments need to be performed by, for instance, employing a vesicular membrane preparation that contains reconstituted DAT or NET. Such preparations allow for the control of the inner and outer ion composition while preventing the substrate to escape from the vesicular confinement. However, steady-state assessment of transporter-mediated substrate uptake is hindered by the fact that all three monoamine transporters can also transport substrate in the absence of $K_{in}^+$. These observations are difficult to reconcile with the concept of transport by fixed stoichiometry. We, therefore, surmise that DAT, NET, and SERT operate with a mixed stoichiometry. Based on our data we conclude that DAT and NET are less likely than SERT to antiport $K_{in}^+$, because we cannot rule out that they can occasionally carry the $K_{in}^+$ ion through the membrane. Conversely, SERT antiports $K_{in}^+$ in the majority of its cycles but may return empty in some instances. We thus believe that the differences between these three transporters with respect to their handling of $K_{in}^+$ represent a continuum, as opposed to divergence, in ionic coupling and kinetic decision points during substrate transport. In this context, it is worth pointing out that the family of *SLC1* glutamate transporters, another important family of neurotransmitter transporters, comprises members that are categorized as either (i) $K^+$-dependent, that is, they utilize $K_{in}^+$ as an obligatory co-substrate or (ii) $K^+$-independent, that is, they do not require $K_{in}^+$ for completion of the transport cycle (*Alleva and Machtens, 2021*). However, $K_{in}^+$ was shown to bind to the same binding sites in both subfamilies (*Wang et al., 2019*; *Kortzak et al., 2019*; *Wang et al., 2020*). We note that the presence of $K_{in}^+$ is not obligatory for any representative of the monoamine transporters. Even SERT, which antiports $K_{in}^+$, can transport substrate in the absence of $K^+$; this

occurs at a lower velocity (*Figure 6G* and *Hasenhuetl et al., 2016*). It is not clear whether $K_{in}^+$-independent *SLC1* transporters can still utilize the $K^+$ gradient to fuel concentrative uptake. If these transporters do, they would mechanistically resemble SERT, if not we would expect them to operate like DAT and NET.

The differences in $K_{in}^+$ handling by SERT and by DAT and NET and concomitant differences in voltage-dependence profiles may have a distinct physiological impact and may have arisen due to evolutionary optimization and adaptation to physiological requirements. An analysis based on the alignment of the primary sequences of DAT, NET, and SERT reveals that SERT is the most divergent member of the three monoamine transporters. SERT is expressed, in both neurons and non-excitable cells, for example, in the trophoblast of the placenta (*Balkovetz et al., 1989*; *Kliman et al., 2018*; *Rudnick and Sandtner, 2019*). The membrane potential in these cells can range from at least 0 mV to −84 mV, with further variations under different biological processes (*Carstensen et al., 1973*; *Friedhoff and Sonenberg, 1983*; *Greenwood et al., 1996*; *Birdsey et al., 1997*; *Kirby et al., 2003*). Thus, SERT must operate under very different transmembrane voltages. It is, therefore, conceivable that the voltage independence of SERT is an evolutionary adaptation to maintain effective forward transport over a large variation in membrane potential. DAT, on the other hand, has been previously predicted to exploit membrane hyperpolarization caused by activated autoreceptors and open G protein-coupled inwardly rectifying potassium channel to augment dopamine reuptake and synaptic clearance (*Sonders et al., 1997*). Hence, there seems to be a trade-off in plasmalemmal monoamine transporter function. Because of electrogenic binding and subsequent counter-transport of $K_{in}^+$, SERT operates in the forward transport mode with a constant rate regardless of membrane potential, but it cannot exploit the membrane potential to fuel its concentrative power. In contrast, DAT and NET can harvest the energy of the transmembrane potential to fuel its concentrative power. As a trade-off, the substrate uptake rate of DAT and NET is voltage-dependent and strongly reduced or increased upon membrane depolarization or hyperpolarization, respectively.

## Materials and methods

### Whole-cell patch clamping

Whole-cell patch-clamp experiments were performed on HEK293 cells stably expressing DAT, NET, or SERT. HEK293 cells, previously purchased from ATCC (# CRL-1573; ATCC, USA), were authenticated by STR profiling at the Medical University of Graz (Cell Culture Core Facility). These cells were then used in control experiments or to generate stable lines expressing either SERT, NET, or DAT. The cells were regularly tested for mycoplasma contamination by 4′,6-diamidino-2-phenylindole staining. These cells were grown in Dulbecco's Modified Eagle Media (DMEM) supplemented with 10% heat-inactivated fetal calf serum (FBS), 100 u·100 ml$^{-1}$ penicillin, 100 u·100 ml$^{-1}$ streptomycin, and 100 μg ml$^{-1}$ of geneticin/G418 for positive selection of transporter-expressing clones. None of the cell lines used in this study were included in the list of commonly misidentified cell lines maintained by International Cell Line Authentication Committee.

24 hr prior to patching, the cells were seeded at a low density on PDL-coated 35 mm plates. Substrate-induced transporter currents were recorded under voltage clamp. Cells were continuously superfused with a physiological external solution that contains 163 mM NaCl, 2.5 mM CaCl$_2$, 2 mM MgCl$_2$, 20 mM glucose, and 10 mM 4-(2-hydroxyethyl)-1-piperazineethanesulfonic acid (HEPES) (pH adjusted to 7.4 with NaOH). A pipette solution mimicking the internal ionic composition of a cell (referred to as normal internal solution henceforth) contained 133 mM potassium gluconate, 6 mM NaCl, 1 mM CaCl$_2$, 0.7 mM MgCl$_2$, 10 mM HEPES, and 10 mM ethylene glycol tetraacetic acid (EGTA) (pH adjusted to 7.2 with KOH, final $K_{in}^+$ concentration - 163 mM). A low-Cl$^-$ internal solution was made by replacing NaCl, CaCl$_2$, and MgCl$_2$ in normal internal solution by NaMES, CaMES, and Mg-acetate (MES - methanesulfonate). A high-Cl$^-$ internal solution was made by replacing potassium gluconate in the normal internal solution with KCl. $Na_{in}^+$ and/or $K_{in}^+$-free internal solutions were made by replacing NaCl and/or potassium gluconate, respectively, in the normal internal solution with equimolar concentrations of NMDG chloride (titrated to a pH of 7.2 using either NMDG or KOH in $Na_{in}^+$-free 163 mM $K_{in}^+$ internal solution). An internal solution with a high $Na_{in}^+$ concentration was made by replacing potassium gluconate of the normal internal solution with an equimolar concentration of NaCl (pH adjusted to 7.2 with NaOH). A high-Li$^+$ internal solution was made by

replacing potassium gluconate in the normal internal solution with 130 mM of LiCl (pH adjusted to 7.2 with LiOH, final $Li_{in}^+$ concentration - 163 mM). An internal solution with a pH of 5.6 was prepared with 10 mM 2-(N-morpholino)ethanesulfonic acid buffer, 1 mM $CaCl_2$, 0.7 mM $MgCl_2$, 10 mM EGTA, and 140 mM NMDGCl and was titrated to pH 5.6 with NMDG. Currents elicited by dopamine or $APP^+$, a fluorescent substrate of DAT (IDT307; Sigma Aldrich), were measured at room temperature (20–24°C) using an Axopatch 200B amplifier and pClamp 10.2 software (MDS Analytical Technologies). Dopamine, norepinephrine, serotonin, or $APP^+$ was applied using a DAD-12 superfusion system and a four-tube perfusion manifold (ALA Scientific Instruments), which allowed for rapid solution exchange. Current traces were filtered at 1 kHz and digitized at 10 kHz using Digidata 1550 (MDS Analytical Technologies). Current amplitudes and accompanying kinetics in response to substrate application were quantified using Clampfit 10.2 software (Molecular Devices). Passive holding currents were subtracted, and the traces were filtered using a 100 Hz digital Gaussian low-pass filter.

## Simultaneous fluorescence-current recordings and epifluorescence imaging

24 hr prior to fluorescence recording and epifluorescence imaging, either untransfected HEK293 cells or HEK293 cells stably expressing DAT, NET, or SERT were seeded at a low density on PDL-coated 35 mm glass bottom dishes, which have a cover glass (Cellview Cell Culture Dish; Greiner Bio-One GmbH, Germany). On the day of the experiment for fluorescence recordings, individual cells were visualized and patched using a x100 oil-immersion objective under voltage clamp. $APP^+$, a fluorescent molecule that has an excitation range from 420 to 450 nm, was applied to single cells using a perfusion manifold. $APP^+$ uptake into the cell was measured using an LED lamp emitting 440 nm light and a dichroic mirror that reflected the light onto the cells. The emitted fluorescence from the sequestered $APP^+$ within the cells was measured using photomultiplier tubes (PMT2102; Thorlabs, United States) mounted on the microscope after it had passed an emission filter. The signal from the PMT was filtered at 3 kHz and digitized at 10 kHz with Axon Digidata 1550B and pClamp 10.2 software (MDS Analytical Technologies). Current traces were filtered as mentioned above. The signals (i.e., currents and fluorescence) were acquired with separate channels. For the epifluorescence imaging, $APP^+$ fluorescence was excited using a 435 nm LED illumination device (Cooled pE4000) and the optical filters described above. Fluorescence images were taken every 250 ms at a resolution of 512 × 512 px on an sCMOS camera (Andor Zyla 5.5, 12 bit mode, readout speed 200 MHz, conversion gain 3, 100 ms exposure time) using a Nikon x40 water immersion (NA 1.25) objective on a Nikon Eclipse Ti2 inverted microscope. $APP^+$ was applied using a four-tube perfusion manifold (ALA Scientific Instruments).

## Kinetic modeling and statistics

The kinetic model for the DAT transport cycle was based on previously reported sequential binding models for DAT (*Erreger et al., 2008*) and SERT (*Hasenhuetl et al., 2016*). State occupancies were calculated by numerical integration of the resulting system of differential equations using Systems Biology Toolbox (*Schmidt and Jirstrand, 2006*) and MAT LAB 2017a software (Mathworks). The voltage dependence of individual partial reactions was modeled assuming a symmetric barrier as $k_{ij} = k^0_{ij}e^{-zQ_{ij}FV/2RT}$, where F = 96,485 C·mol$^{-1}$, R = 8.314 JK$^{-1}$mol$^{-1}$, V is the membrane voltage in volts, and T = 293 K (*Läuger, 1991*). Coupled membrane currents upon application of substrate were calculated as I = $-F \times NC/N_A \times \Sigma z_{Qij}(p_ik_{ij} - p_jk_{ji})$, where $z_{Qij}$ is the net charge transferred during the transition, NC is the number of transporters ($4 \times 10^6$/cell), and NA = $6.022e^{23}$/mol. The substrate-induced uncoupled current was modeled as a current through a $Na^+$-permeable channel with I = $P_o\gamma NC(V_M - V_{rev})$, where $P_o$ corresponds to the occupancy of the channel state, $\gamma$ is the single-channel conductance of 2.4 pS, $V_M$ is the membrane voltage, and $V_{rev}$ is the reversal potential of $Na^+$ (+100 mV). The extracellular and intracellular ion concentrations were set to the values used in the respective experiments. To account for the non-instantaneous onset of the substrate in patch-clamp experiments, we modeled the substrate application as an exponential rise with a time constant of 10 ms. Uptake of $APP^+$ was modeled as $TiClS*k_{off}S_{in} - TiCl*k_{on}S_{in}*S_{in}* NC/N_A$, where TiClS and TiCl are the respective state occupancies, $k_{on}S_{in}$ and $k_{off}S_{in}$ are the association and dissociation rate constants of $APP^+$, and $S_{in}$ is the intracellular concentration of $APP^+$. Experimental variations are either reported as means ± 95% confidence intervals, means ± SD, or means ± SEM. Some of the

data were fit to the Boltzmann equation (Y = Bottom + (Top-Bottom)/(1 + exp((V50-X)/Slope)) or the line function (linear regression)). However, both were arbitrary fits to the data. Neither one of them were suitable to model the processes, which underlie the depicted voltage dependence. The decision to use one or the other was based on the fidelity of the resulting fit.

## Acknowledgements

We thank Verena Burtscher for discussion and comments on the data.

## Additional information

### Funding

| Funder | Grant reference number | Author |
|---|---|---|
| Austrian Science Fund | P31599 | Walter Sandtner |
| Austrian Science Fund | P31813 | Walter Sandtner |
| Austrian Science Fund | W1232 | Harald H Sitte |
| Vienna Science and Technology Fund | CS15-033 | Harald H Sitte |
| Vienna Science and Technology Fund | LS17-026 | Michael Freissmuth |

The funders had no role in study design, data collection and interpretation, or the decision to submit the work for publication.

### Author contributions

Shreyas Bhat, Conceptualization, Data curation, Formal analysis, Investigation, Methodology, Writing - original draft, Writing - review and editing; Marco Niello, Conceptualization, Data curation, Formal analysis, Investigation, Writing - original draft; Klaus Schicker, Conceptualization, Formal analysis, Methodology, Writing - review and editing; Christian Pifl, Conceptualization, Resources, Writing - review and editing; Harald H Sitte, Resources, Supervision, Funding acquisition, Writing - review and editing; Michael Freissmuth, Conceptualization, Resources, Supervision, Funding acquisition, Writing - review and editing; Walter Sandtner, Conceptualization, Resources, Formal analysis, Supervision, Funding acquisition, Methodology, Writing - original draft, Project administration, Writing - review and editing

### Author ORCIDs

Shreyas Bhat (ID) https://orcid.org/0000-0001-7019-9180
Harald H Sitte (ID) http://orcid.org/0000-0002-1339-7444
Walter Sandtner (ID) https://orcid.org/0000-0003-3637-260X

### Decision letter and Author response

Decision letter https://doi.org/10.7554/eLife.67996.sa1
Author response https://doi.org/10.7554/eLife.67996.sa2

## Additional files

### Supplementary files

• Transparent reporting form

### Data availability

All data generated or analysed during this study are included in the manuscript and supporting files. Source data files have been provided for Figures 2, 3, 5, and 6 in the following DOI published by dryad. https://doi.org/10.5061/dryad.6q573n5z8.

The following dataset was generated:

| Author(s) | Year | Dataset title | Dataset URL | Database and Identifier |
|---|---|---|---|---|
| Sandtner W | 2021 | | https://doi.org/10.5061/dryad.6q573n5z8 | Dryad Digital Repository, 10.5061/dryad.6q573n5z8 |

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
