## [Decision Letter]

**Acceptance summary:**

This article reports an insightful, quantitative study of an important class of ion-coupled membrane transporters. The authors demonstrate that dopamine, catecholamine and serotonin transporters – albeit structurally very similar – differ in the number of transported substrates; the authors also make good progress towards defining the underlying basis of this differentiation, using a range of sophisticated experimental techniques. Since effective pharmaceutical approaches often require that drugs target one transporter but not others, molecular-level investigations of the factors that explain the emergence of functional specificity are not only significant from a fundamental standpoint but also from a biomedical perspective.

**Decision letter after peer review:**

Thank you for submitting your article "Handling of intracellular K + determines voltage dependence of plasmalemmal monoamine transporter function" for consideration by *eLife*. Your article has been reviewed by 3 peer reviewers, one of whom is a member of our Board of Reviewing Editors, and the evaluation has been overseen by José Faraldo-Gómez as the Senior Editor. Only reviewer #3 has agreed to reveal his identity (Baruch Kanner).

*Reviewer #1 (Recommendations for the authors):*

The authors should address several concerns:

1. The authors suggest that APP+ binds to the membrane, resulting in the unspecific response in non-transfected cells. This should be validated by another method, for example fluorescence microscopy.

2. Why doesn't APP+ generate a fluorescent signal when it flows out of the solution exchange device? In fact, Figure 1A shows the APP+ solution as colorless before it enters the cell. Why is this the case? Wouldn't it be more likely that APP+ is also fluorescent in the superfusate? If the latter is the case, does this result in a large background in the fluorescent signal?

3. The APP+ dose response curve needs to be shown for the non-specific signal in control, non-transfected cells.

4. Once APP+ enters the cell, why is it not diluted by diffusion into the whole-cell recording pipet? In fact, accumulation in the cytosol, despite diffusive equilibration with the pipet solution would indicate slow diffusion. On the other hand, slow diffusion and APP+ accumulation would mean that trans inhibition could result in a reduction of the whole-cell current over time. However, such a reduction is not seen. Therefore, the APP+ fluorescence and current recording experiments are somewhat at odds with each other.

5. The point in the last paragraph could be experimentally tested using the perforated patch whole-cell recording method, which could prevent diffusion of APP+ into the pipet.

6. Isn't it possible that the reduction of DAT steady state current in the absence of K^+^ is caused by a regulatory K^+^ binding site? Why does the K^+^ binding and dissociation process have to be included in the actual transport cycle?

7. If the DAT dopamine steady state current is reduced in the absence of intracellular K^+^, why is the same not the case for APP+ uptake (fluorescence), which is also at steady state?

*Reviewer #2 (Recommendations for the authors):*

1. Since the authors speculate about "DAT, NET and SERT operat(ing) with a mixed stoichiometry", I wonder whether one might be able to assess the transport stoichiometry from measurements that combine fluorescence time course, current amplitudes and cell sizes. The authors might want to comment on this issue.

2. In the discussion, the authors nicely explain their findings in the context of transport properties. However, a potential physiological impact of the differences in voltage dependence of DAT, NET and SERT is lacking. In light of the fierce evolutionary optimization our transport proteins have gone through one would expect a benefit of voltage-independent serotonin uptake and of voltage-dependent dopamine and catecholamine uptake. The authors might want to speculate about this.

3. The two most important neurotransmitter transporter families, the SLC1 and the SLC6 family, encompass K^+^-dependent and K^+^-independent isoforms. The authors might want to compare mechanisms and physiological impacts of the variable K^+^ coupling in such transporters.

*Reviewer #3 (Recommendations for the authors):*

1. Symbols in the Figures are too small.

2. Explain in the first section of the Results which processes the peak and steady state currents reflect.

3. Give some background on the known structures for the readers: what is Na_2_ and what is its importance in the translocation cycle.

4. Bottom of p. 5, alternative explanation at this point of the story (Figures 1 and 2): NET and SERT are very similar but the latter has a large uncoupled current.

5. Bottom of p.7: voltage dependence of transport can be in theory due to the effect of voltage of a rate-limiting step of an electroneutral transport cycle. This should be mentioned.

6. Legend to Figure 4: what is the substitution cation? NMDG+?

7. Need the li control also in Figures 5Kand L. What are the actual (not normalized) peak currents at -60 mV under the various "in" conditions in each of the panels J, K and L?

8. Figure 7: "Scheme" rather than "Schema" (which is probably the expression in German).

9. Something is wrong with the ionic conditions on lines 3 and 4 of the legend to Figure S1.

---

## [Author Response]

Reviewer #1 (Recommendations for the authors):The authors should address several concerns:1. The authors suggest that APP+ binds to the membrane, resulting in the unspecific response in non-transfected cells. This should be validated by another method, for example fluorescence microscopy.

We thank the reviewer for this comment. As recommended, we used a camera to obtain spatial information. These data indicate that the fast component in the fluorescence signal is due to photons emitted by the low quantum yield state, which APP^+^ adopts when present in a polar solvent. We show these data in the new supplementary figure 1. Accordingly our initial conjecture that the fast component reflects APP^+^ binding to the plasma membrane was false. Please refer to the next response for more details.

2. Why doesn't APP+ generate a fluorescent signal when it flows out of the solution exchange device? In fact, Figure 1A shows the APP+ solution as colorless before it enters the cell. Why is this the case? Wouldn't it be more likely that APP+ is also fluorescent in the superfusate? If the latter is the case, does this result in a large background in the fluorescent signal?

We appreciate the reviewer’s perspective on the initial interpretation of our data. We employed PMT (photomuliplier tube) recordings rather than any other fluorescence recording technique, because PMT recordings provide unmatched temporal resolution for the study of the transport cycle of SERT, DAT and NET. However, this approach does not provide any spatial resolution. We erroneously attributed the non-specific fast component to plasma membrane binding of APP^+^ on the basis of a previous report (Karpowicz Jr. et al., 2013, cited in the manuscript): this paper highlights the fact that the APP^+^ fluorescence in polar solvents is significantly lower than in organic solvents. Based on the recommendation of reviewer 1, we employed epifluorescence imaging to examine our previous assumption about the fast non-specific component. As it turned out, reviewer 1’s hunch was right: APP^+^ showed background fluorescence during solution exchange. This fluorescence was, however, faint when compared to APP^+^ fluorescence arising due to its uptake in DAT expressing cells (supplementary figure 1).

Accordingly, we have now incorporated the new interpretation of the fast component of fluorescence rise as follows: (a) *Figure 1B*, APP^+^ is light orange in bath solution but increases in intensity (dark orange) once present inside the cell and (b) in the manuscript text, we have added the following sentences (lines 125-138):

“The fluorescence intensity of pyridinium dyes such as APP^+^ is much lower in polar solvents than in their hydrophobic counterparts. […] Thus, although the fluorescence of APP^+^ in solution is low, its integration over a large area amounted to the rapidly rising and declining component, which we also observed when using the PMT.”

3. The APP+ dose response curve needs to be shown for the non-specific signal in control, non-transfected cells.

We have now incorporated APP^+^ concentration response curve in control cells as *supplementary figure 2*.

4. Once APP+ enters the cell, why is it not diluted by diffusion into the whole-cell recording pipet? In fact, accumulation in the cytosol, despite diffusive equilibration with the pipet solution would indicate slow diffusion. On the other hand, slow diffusion and APP+ accumulation would mean that trans inhibition could result in a reduction of the whole-cell current over time. However, such a reduction is not seen. Therefore, the APP+ fluorescence and current recording experiments are somewhat at odds with each other.

We show by epifluorescence imaging that upon entering into the cell APP^+^ adheres predominantly to membranes of intracellular organelles (supplementary figure 1). This has also been previously observed by others (Solis Jr. et al., 2012; Karpowicz Jr et al., 2013; Wilson et al., 2014, cited in the manuscript). Accordingly, we believe that there is little free cytosolic APP^+^ and as a consequence even less APP^+^ entering into the pipette.

5. The point in the last paragraph could be experimentally tested using the perforated patch whole-cell recording method, which could prevent diffusion of APP+ into the pipet.

As pointed out in the reply to point 4, binding of APP+ to intracellular membranes (which we visualized, see above) precludes the accumulation of free cytosolic APP^+^. Hence, diffusion into the pipette is – in our opinion, negligible, at least not in the time frame of our recordings. In addition, we refrained from recordings in the perforated patch configuration, because this method does not allow for the required ionic exchange (e.g. full removal of intracellular K^+^).

6. Isn't it possible that the reduction of DAT steady state current in the absence of K^+^ is caused by a regulatory K^+^ binding site? Why does the K^+^ binding and dissociation process have to be included in the actual transport cycle?

Please refer to our response to the public review for reviewer 1, where we provide our arguments against a regulatory K^+^ binding site in DAT.

7. If the DAT dopamine steady state current is reduced in the absence of intracellular K^+^, why is the same not the case for APP+ uptake (fluorescence), which is also at steady state?

By definition, the linear increase in APP^+^ fluorescence is proportional to DAT mediated APP^+^ uptake. In the presence of intracellular K^+^, DAT mediated currents are comprised of two components: one that is coupled to substrate uptake and one that is not. Accordingly, removal of intracellular K^+^ abolishes the uncoupled component. This can only be seen only in DAT currents (but not in APP+ uptake, which corresponds to the coupled component).

Reviewer #2 (Recommendations for the authors):1. Since the authors speculate about "DAT, NET and SERT operat(ing) with a mixed stoichiometry", I wonder whether one might be able to assess the transport stoichiometry from measurements that combine fluorescence time course, current amplitudes and cell sizes. The authors might want to comment on this issue.

Knowledge of transport kinetics (i.e. measurements of time courses) does not provide definitive insights into the stoichiometry of a transporter. This assessment requires measurements of the concentrative power of a transporter at equilibrium. One disadvantage of whole cell patch clamp recordings is that such an equilibrium cannot be reached (due to escape of the substrate into the patch electrode). In addition, the cytosolic concentration of APP^+^ is mainly determined by its binding capacity to the intracellular structures and not by the pertinent transmembrane ion gradients or voltage. Because of this binding, the cytosolic concentration of APP^+^ does not reach sufficient levels within a plausible time frame to allow for exchange. Hence, we believe that the methods used in our study are not suitable to address transport stoichiometry in a rigorous manner.

2. In the discussion, the authors nicely explain their findings in the context of transport properties. However, a potential physiological impact of the differences in voltage dependence of DAT, NET and SERT is lacking. In light of the fierce evolutionary optimization our transport proteins have gone through one would expect a benefit of voltage-independent serotonin uptake and of voltage-dependent dopamine and catecholamine uptake. The authors might want to speculate about this.

We have followed up on the reviewer’s suggestion by incorporating several sentences in the last paragraph of the discussion. These read as follows (lines 625-648):

“The differences in K_in_^+^ handling by SERT and by DAT and NET and concomitant differences in voltage-dependence profiles may have a distinct physiological impact and may have arisen due to evolutionary optimization and adaptation to physiological requirements. […] As a tradeoff, the substrate uptake rate of DAT and NET is voltage-dependent and strongly reduced or increased upon membrane depolarization or hyperpolarization, respectively.”

3. The two most important neurotransmitter transporter families, the SLC1 and the SLC6 family, encompass K^+^-dependent and K^+^-independent isoforms. The authors might want to compare mechanisms and physiological impacts of the variable K^+^ coupling in such transporters.

We thank the reviewer for this suggestion since *SLC1* transporters encompass members that are either K^+^-dependent or K^+^-independent isoforms. We addressed this in the discussion as follows: (lines 613624):

“In this context, it is worth pointing out that the family of *SLC1* glutamate transporters, another important family of neurotransmitter transporters, is comprised of members, which are categorized as (i) K^+^-dependent, i.e., they utilize K_in_^+^ as an obligatory co-substrate or (ii) K^+^-independent, i.e., they do not require K_in_^+^ for completion of the transport cycle (Alleva et al., 2021). […] If these transporters do, they would mechanistically resemble SERT; if not, we would expect them to operate like DAT and NET.”

Reviewer #3 (Recommendations for the authors):1. Symbols in the Figures are too small.

We sincerely apologise for this inconvenience. The figure symbols have been enlarged for ease in visualization of the figures.

2. Explain in the first section of the Results which processes the peak and steady state currents reflect.

We thank Dr. Kanner for this suggestion. We have added the following text in the manuscript to explain the individual components of substrate-induced transporter-mediated currents (lines 99106):

“The scheme in Figure 1C is a simplified representation of a transport cycle endured by a sodium and chloride coupled secondary active transporter. […] Accordingly, only substrates can induce steady state currents while inhibitors cannot.”

In addition, we have included a simplistic scheme of the transport cycle that sodium and chloride coupled secondary active transporters undergo as Figure 1C. We highlighted the reactions that give rise to the peak and steady state components of substrate induced transporter mediated currents in Figure 1B, right and Figure 1C.

3. Give some background on the known structures for the readers: what is Na_2_ and what is its importance in the translocation cycle.

We have added the following text to explain the importance of the Na_2_ site and its relevance to our experimental observations (lines 510-524):

“The binding site for K_in_^+^ in DAT, NET and SERT is largely unknown. Binding of K_in_^+^ to DAT at the Na_2_ site was proposed earlier by a study that employed extensive molecular dynamic simulations to understand intracellular Na^+^ dissociation from DAT (Razavi et al., 2017). […] Our results highlight the fact that the voltage dependence of peak amplitudes is identical in the presence of high Na_in_^+^ and high K_in_^+^ (Figure 5J-L) in all three plasmalemmal monoamine transporters; these observations support the conjecture that K_in_^+^ and Na_in_^+^ binding occurs at the Na_2_ site in a mutually exclusive manner.”

4. Bottom of p. 5, alternative explanation at this point of the story (Figures 1 and 2): NET and SERT are very similar but the latter has a large uncoupled current.

We respectfully disagree with Dr. Kanner’s interpretation of SERT and NET data from Figure 1 and Figure 2. As shown in Figure 6E-G, the catalytic rate of NET is 10-20 fold slower than SERT/DAT (as determined by the “two pulse protocol”, Figure 6D).

5. Bottom of p.7: voltage dependence of transport can be in theory due to the effect of voltage of a rate-limiting step of an electroneutral transport cycle. This should be mentioned.

We have incorporated Dr. Kanner’s recommendation in the discussion. The pertinent sentences (lines 538-541) read as follows:

“We note that the lack of voltage dependence, which we observed in the SERT mediated uptake, is not contingent on its assumed electroneutrality. Even an electroneutral transporter can support voltage-dependent uptake if the rate limiting step is associated with charge movement.”

6. Legend to Figure 4: what is the substitution cation? NMDG+?

We have mentioned the use of NMDG^+^ for cationic substitution in internal solutions that are devoid of Na_in_^+^ and K_in_^+^ in the material and methods section. For the sake of clarity, we have now substituted “0 mM K_in_^+^/Na_in_^+^” with “163 mM NMDG_in_^+^” throughout the manuscript.

7. Need the li control also in Figures 5Kand L. What are the actual (not normalized) peak currents at -60 mV under the various "in" conditions in each of the panels J, K and L?

We thank Dr. Kanner for pointing out this discrepancy in our data representation across Figure 5JL. We have now included Li_in_^+^ data for NET and SERT in Figure 5K and Figure 5L. As expected NET showed Li^+^ profiles similar to those of DAT. Surprisingly, SERT showed a shallow IV profile with Li^+^ in a manner consistent with those of high intracellular K^+^ or Na^+^. This indicates binding of intracellular Li^+^ to SERT. We believe that any further characterization of this interesting phenomena warrants a separate study and is beyond the scope of this manuscript. Please refer to the response to the public review by Dr. Kanner that mentions changes incorporated in the manuscript in support of this newly acquired datasets.

We have, in addition, included the actual peak current amplitudes under different intracellular conditions for DAT, NET and SERT as supplementary figure 3.

8. Figure 7: "Scheme" rather than "Schema" (which is probably the expression in German).

The necessary changes have been incorporated in Figure 7.

9. Something is wrong with the ionic conditions on lines 3 and 4 of the legend to Figure S1.

We apologise for this textual error. We have replaced the erroneous ionic conditions mentioned in supplementary figure 4 (old supplementary figure 1) with 163 mM NMDG_in_^+^.